# ALS-related FUS mutations alter axon growth in motoneurons and affect HuD/ELAVL4 and FMRP activity

Maria Giovanna Garone [1], Nicol Birsa[2,3], Maria Rosito[4], Federico Salaris[1,4], Michela Mochi[1], Valeria de Turris [4], Remya R. Nair [5], Thomas J. Cunningham[5], Elizabeth M. C. Fisher[2], Mariangela Morlando[6], Pietro Fratta [2] & Alessandro Rosa [1,4,7✉]

Mutations in the RNA-binding protein (RBP) FUS have been genetically associated with the motoneuron disease amyotrophic lateral sclerosis (ALS). Using both human induced pluripotent stem cells and mouse models, we found that FUS-ALS causative mutations affect the activity of two relevant RBPs with important roles in neuronal RNA metabolism: HuD/ELAVL4 and FMRP. Mechanistically, mutant FUS leads to upregulation of HuD protein levels through competition with FMRP for *HuD* mRNA 3'UTR binding. In turn, increased HuD levels overly stabilize the transcript levels of its targets, NRN1 and GAP43. As a consequence, mutant FUS motoneurons show increased axon branching and growth upon injury, which could be rescued by dampening NRN1 levels. Since similar phenotypes have been previously described in SOD1 and TDP-43 mutant models, increased axonal growth and branching might represent broad early events in the pathogenesis of ALS.

[1] Department of Biology and Biotechnologies "Charles Darwin", Sapienza University of Rome, Rome, Italy. [2] UCL Queen Square Institute of Neurology, University College London, London, UK. [3] UK Dementia Research Institute, University College London, London, UK. [4] Center for Life Nano- & Neuro-Science, Fondazione Istituto Italiano di Tecnologia (IIT), Rome, Italy. [5] MRC Harwell Institute, Didcot, UK. [6] Department of Pharmaceutical Sciences, "Department of Excellence 2018-2022", University of Perugia, Perugia, Italy. [7] Laboratory Affiliated to Istituto Pasteur Italia-Fondazione Cenci Bolognetti, Department of Biology and Biotechnologies "Charles Darwin", Sapienza University of Rome, Rome, Italy. ✉email: alessandro.rosa@uniroma1.it

The motoneuron disease amyotrophic lateral sclerosis (ALS) has been linked to mutations in several RNA binding proteins (RBPs), including the FUS gene, and altered RNA metabolism[1]. Despite a recent increase in our knowledge of the genetics of ALS, the disease mechanisms downstream of mutations in ALS-genes remain largely uncharacterized. In the RBP FUS the most severe ALS mutations, including the P525L, lie within its C-terminal nuclear localization signal (PY-NLS domain), impairing the interaction with the nuclear import receptor Transportin-1 (TNPO1), and reducing nuclear localization[2]. Loss of ALS-linked RBPs nuclear functions, including regulation of alternative splicing and polyadenylation, has been proposed as a pathological ALS mechanism[3–5]. Insoluble cytoplasmatic aggregates containing ALS-linked RBPs are a hallmark of the pathology[6], and gain of toxic cytoplasmic functions may also play important roles in ALS[7].

We previously observed strong correlation between changes in protein levels and selective binding of mutant FUS to 3′UTR[8], suggesting that aberrant targeting of 3′UTRs by mutant FUS likely represents a broad mechanism underlying proteome alteration in motoneurons (MNs). This is particularly relevant for ALS-linked genes, genes encoding for cytoskeletal proteins and, notably, other RBPs[8,9]. Importantly, cellular levels of ALS-linked RBPs are tightly regulated by both auto-regulation mechanisms[10–12] and cross-regulatory mechanisms[9,12–16]. The neural RBP HuD (encoded by the ELAVL4 gene), a component of cytoplasmic inclusions in FUS, TDP-43, and sporadic ALS patients[9,14], represents an example of such cross-regulation. We have found aberrantly increased HuD levels in FUS mutant MNs due to microRNA-mediated effects, and direct binding of mutant FUS to the HuD 3′UTR, by a still uncharacterized mechanism[9,17]. HuD is a neural multifunctional RBP and its overexpression induces increased neurite outgrowth in neuronal cell lines and primary neural progenitor cells[18–20], but whether increased levels of HuD have functional consequences in FUS mutant MNs remains unexplored.

Axonal degeneration is a key feature in the ALS pathophysiology and occurs prior to the motor phenotype in patients[21–23]. Despite the underlying pathological mechanisms have not been fully elucidated, it has been proposed that axonal alteration, including aberrantly increased branching, can act as a trigger[24]. The levels of cytoskeletal proteins and factors directing neuron projection are changed in FUS mutant human pluripotent stem cell (hiPSC)-derived MNs[8], and aberrantly increased branching and axonal outgrowth have been recently identified across ALS mutations and model systems, underlying their importance in early disease pathogenesis[24–28]. Spinal MNs isolated from adult SOD1 mutant mice at pre-symptomatic stages displayed significant increase in axon outgrowth, in terms of length and branching complexity, and acute expression of mutant SOD1 in WT MNs was sufficient to increase axonal regeneration[29]. These evidences point to axonal alteration as an early, pre-symptomatic phenotype in ALS.

In this work, we aimed to gain insight into the molecular mechanisms leading to HuD upregulation in FUS mutant genetic background, and into the functional consequences of HuD increase in MNs. We provide a mechanistic link between increased axon branching and growth upon axotomy and alteration of a cross-regulatory circuitry involving three RBPs: FUS, HuD and the fragile X mental retardation protein (FMRP). We find that FMRP is a negative regulator of HuD translation via 3′UTR binding. We propose that this function is outcompeted by mutant FUS binding to the same regulatory region, leading to an increase in HuD protein level, thus providing a mechanistic explanation of HuD upregulation in FUS mutant MNs. Further, we identify altered axonal growth as a functional consequence of HuD upregulation, and finally find it to be mediated by the alteration of the HuD target NRN1.

## Results

**ALS mutant FUS competes with FMRP for *HuD* regulation via 3′UTR binding.** We have previously found mutant FUS expression to lead to an increase in HuD[9,17]. Although miR-375 may play a role[17,30], experiments conducted in the absence of miR-375 indicate that a further regulation mechanism is also present. Interestingly, HuD 3′UTR is extensively conserved in vertebrates, with a high phyloP100way score (mean: 4.8; standard deviation: 2.3), approaching that of coding exons (e.g., exon 2, mean: 5.9; standard deviation: 3.2) and not restricted to miR-375 binding sites (Supplementary Fig. 1), further supporting the existence of another regulatory mechanism.

In order to gain insights into HuD regulation in ALS, we took advantage of spinal MNs derived from isogenic pairs of FUS WT and P525L hiPSC lines (hereafter FUS^WT and FUS^P525L) by inducible expression of a "programming module" consisting of the transcription factors Ngn2, Isl1 and Lhx3 (NIL)[31,32]. The P525L mutation, localized in the PY-NLS domain (Supplementary Fig. 2a), causes severe mislocalization of the FUS protein in the cytoplasm and is often associated to juvenile ALS[2]. In parallel, we used the Fus-Δ14 knock-in mouse model, in which a frameshift mutation leads to the loss of the C-terminal nuclear localization signal (NLS) (Supplementary Fig. 2a), causing partial mislocalization to the cytoplasm without altering total Fus protein levels (Supplementary Fig. 2b)[33]. In both human in vitro MNs (Fig. 1a) and mouse spinal cords (post-natal day 81, P81) (Fig. 1b), we observed a two-fold increase of HuD protein levels in FUS mutant genetic backgrounds. Fluorescence in situ hybridization (FISH) analysis showed a significant increase in the number of HuD mRNA puncta in FUS^P525L human MNs (Fig. 1c). We next took advantage of puro-PLA, a technique that couples puromycylation with the proximity-ligation assay to visualize newly synthesized proteins[34]. Puro-PLA was performed on FUS^P525L and FUS^WT human MNs using an anti-HuD antibody, revealing an increase in newly synthesized HuD in mutant cells (Fig. 1d and Supplementary Fig. 3). Increased HuD translation was also detected in primary MNs from the Fus-Δ14 mouse model (Fig. 1e).

Together with our previous work[9], these observations suggest that mutant FUS might trigger HuD translation upregulation via 3′UTR binding. Mechanistically, this effect could arise from the competition with an inhibitory RBP. We used catRAPID[35] to predict the interactors of the HuD 3′UTR. Several RBPs showed interaction propensity with this sequence (Supplementary Data 1), and we filtered this list for ones involved in the regulation of translation (GO:0006412) (Supplementary Table 1). Among these candidates, we focused on negative regulators of translation and noticed FMRP, encoded by the fragile X mental retardation 1 (FMR1) gene. Native RNA immunoprecipitation (RIP) was used to validate the physical association between FMRP and the HuD transcript. FMRP was effectively immunoprecipitated from hiPSC-derived MN extracts (Fig. 2a). A hemizygous FMRP knock-out hiPSC line (FMRP^KO), generated by CRISPR/Cas9-mediated modification of a male line (hereafter FMRP^WT) (Supplementary Fig. 4)[36], served as a negative control. Quantitative RT-PCR analysis revealed specific enrichment of the MAP1B mRNA, a well-characterized FMRP interactor[37,38], and no enrichment of a negative control, the housekeeping ATP5O mRNA, in the FMRP immunoprecipitated samples (Fig. 2b). Consistent with the catRAPID prediction, FMRP immunoprecipitation enriched the HuD mRNA (Fig. 2b). Both MAP1B and HuD were negligible in the immunoprecipitated samples from the

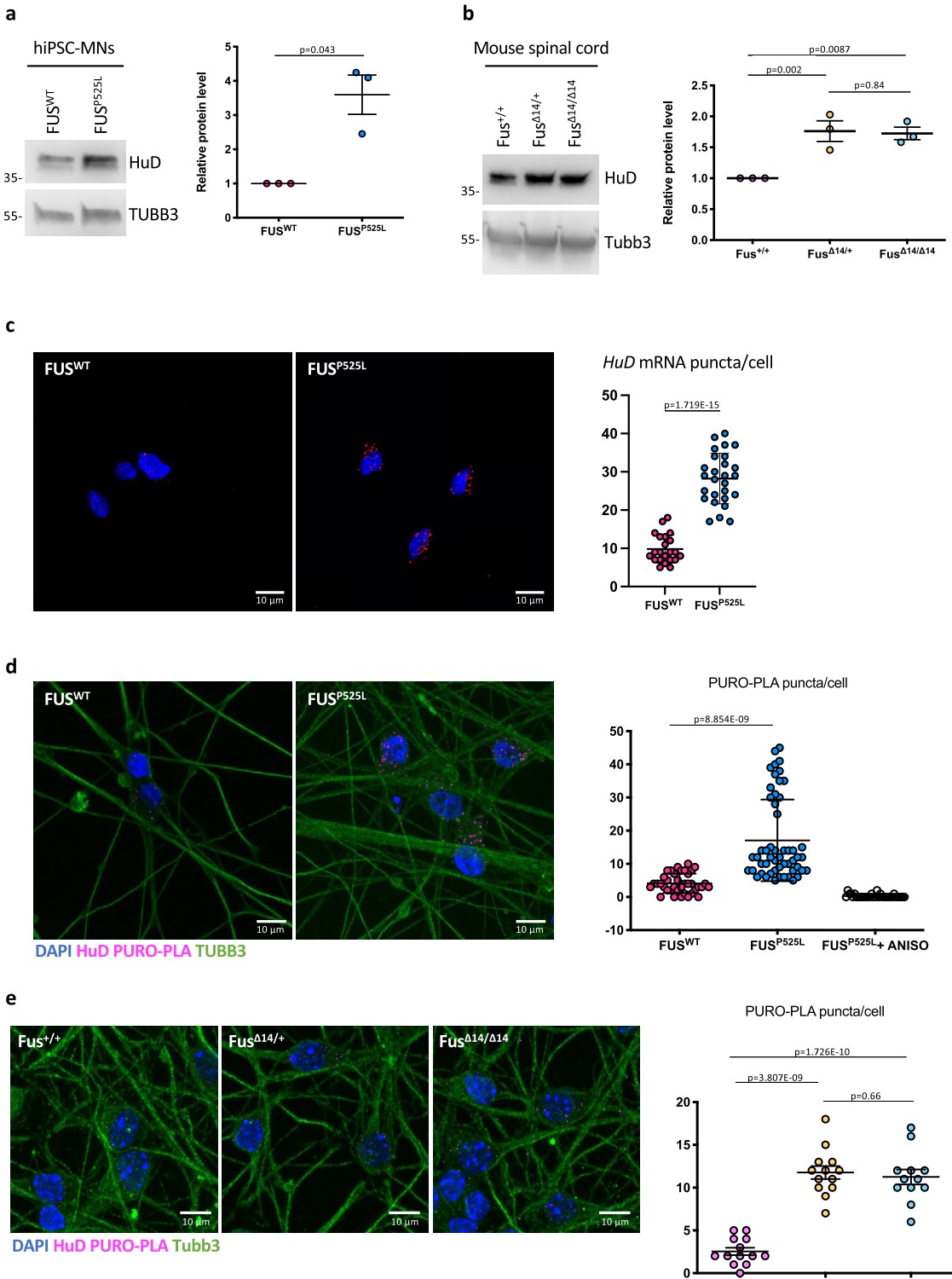

**Fig. 1 Increased HuD levels in human and mouse FUS-mutant MNs. a, b** HuD protein levels analysis by western blot in FUS$^{WT}$ and FUS$^{P525L}$ hiPSC-derived spinal MNs (**a**) and Fus-Δ14 mouse model spinal cord (P81) (**b**). The molecular weight (kDa) is indicated on the left. The graphs show the average from three independent biological replicates, error bars indicate the standard deviation (**a** Student's *t*-test, paired, two tails; **b** ordinary one-way ANOVA, multiple comparisons). TUBB3/Tubb3 signal was used for normalization. Protein levels are relative to FUS$^{WT}$ (**a**) or Fus$^{+/+}$ (**b**) conditions. **c** HuD mRNA analysis by FISH in FUS$^{WT}$ and FUS$^{P525L}$ hiPSC-derived spinal MNs. The graphs show the average count of *HuD* mRNA puncta per cell from three independent differentiation experiments, error bars indicate the standard error of the mean (Student's *t*-test; paired; two tails). **d, e** Combined PURO-PLA (HuD, magenta) and immunostaining (TUBB3, green) analysis in FUS$^{WT}$ and FUS$^{P525L}$ hiPSC-derived spinal MNs (**d**) and primary MNs from Fus-Δ14 mouse embryos (E12.5–13.5) (**e**). DAPI (blue) was used for nuclear staining. **d** FUS$^{P525L}$ hiPSC-derived spinal MNs treated with the eukaryotic protein synthesis inhibitor anisomycin (ANISO) were used as negative control of the PURO-PLA. The graphs show the average count of HuD PURO-PLA puncta per cell from three independent differentiation experiments (**d**) and three samples (**e**), error bars indicate the standard error of the mean (**d** Student's *t*-test, unpaired, two tails; **e** ordinary one-way ANOVA, multiple comparisons). Scale bars (**c–e**): 10 µm.

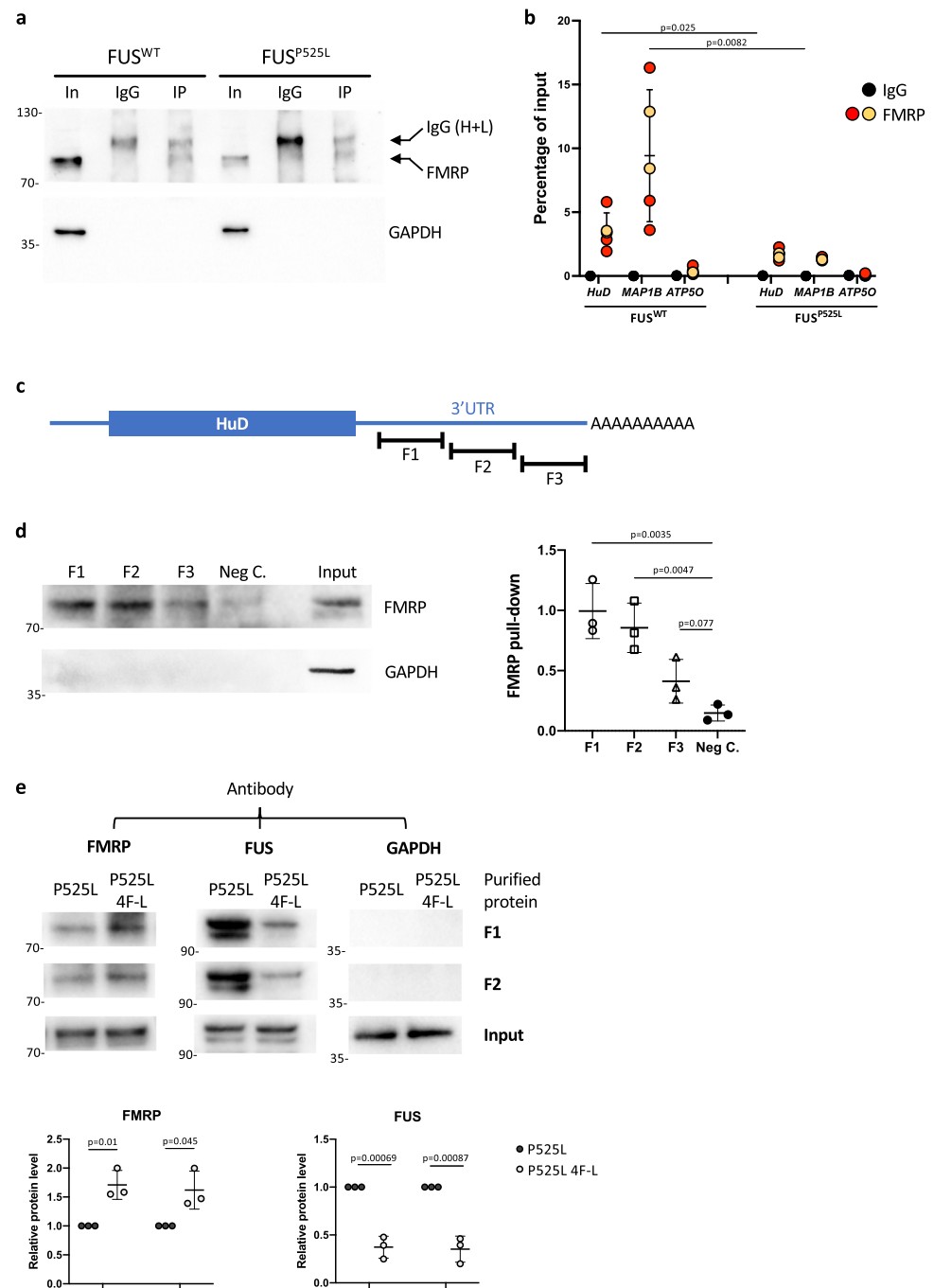

control FMRP[KO] line, confirming the RIP specificity (Supplementary Fig. 4d). Notably, *HuD* mRNA levels were reduced in FMRP immunoprecipitated samples from FUS[P525L] MNs (Fig. 2b). Consistent with a previous report[14], we also observed reduced interaction between FMRP and *MAP1B* mRNA in FUS mutant cells (Fig. 2b). Decreased interaction with its targets was not consequence of lower FMRP protein level in FUS[P525L] MNs (Supplementary Fig. 4e).

In order to directly assess whether FMRP and mutant FUS compete for *HuD* 3′UTR binding, we set up an in vitro binding and competition assay. Three fragments of about 700 nucleotides each, spanning the long *HuD* 3′UTR (F1–F3 in Fig. 2c and Supplementary Fig. 1), along with a negative control (a portion of the Renilla luciferase coding sequence) were in vitro transcribed as biotinylated RNAs and incubated with HeLa cytoplasmic

extract. Upon pull-down with streptavidin beads, western blot analysis revealed strong enrichment of FMRP with F1 and F2, while F3 was not significantly different from the negative control (Fig. 2d). We then repeated this experiment for F1 and F2 in presence of the purified recombinant proteins FUS-P525L (RFP-flag-FUS[P525L], indicated as P525L in Fig. 2e) and FUS-P525L 4F-L, a derivative of the FUS-P525L mutant in which four aminoacidic changes impair its RNA binding ability[39] (P525L 4F-L in Fig. 2e) (Supplementary Fig. 5). For both F1 and F2, we observed strong binding of FUS-P525L, while the RNA-binding defective derivative showed reduced enrichment (Fig. 2e). Conversely, FMRP binding was reduced in presence of RNA-binding competent FUS, compared to the 4F-L derivative (Fig. 2e).

We then aimed to assess the consequences of impaired FMRP binding to *HuD* 3′UTR. In MNs obtained from FMRP[KO] hiPSCs,

**Fig. 2 FMRP and FUS^P525L compete for HuD 3′UTR binding. a** Western blot analysis of the FMRP RIP assay. In input, IgG control immunoprecipitation with rabbit monoclonal anti-human, IgG antibod, IP samples immunoprecipitated with an anti-FMRP antibody. The molecular weight (kDa) is indicated on the left. **b** Analysis of *MAP1B* (positive control), *ATP5O* (negative control) and *HuD* mRNA levels by real time qRT-PCR in samples from FUS^WT or FUS^P525L hiPSC-derived spinal MNs. The graph shows the relative enrichment of the mRNAs pulled down by FMRP, reported as the percentage of input, in IP or control IgG samples, after normalization with an artificial spike RNA. The graphs show the average from five independent differentiation experiments and error bars indicate the standard deviation (Student's *t*-test; unpaired; two tails). For anti-FMRP IP samples, yellow dots are related to samples immunoprecipitated with the ab17722 antibody and orange dots to samples immunoprecipitated with the f4055 antibody. **c** Schematic representation of the HuD transcript. The three regions of the 3′UTR (F1, F2, and F3) used for in vitro binding assays are shown. **d** The in vitro binding assay was performed by incubating biotinylated transcripts corresponding to *HuD* 3′UTR regions F1, F2, or F3, or a portion of the Renilla luciferase coding sequence used as negative control (Neg. C), with HeLa cytoplasmic extract, followed by pull-down with streptavidin-conjugated beads. Western blot analysis was then performed with anti-FMRP antibody to detect FMRP binding. Anti-GAPDH was used as negative control. Input: 10% of the pull-down input sample. The molecular weight (kDa) is indicated on the left. The histogram shows quantification from three independent experiments. Values were calculated as fraction of Input (Student's *t*-test; paired; two tails). **e** The in vitro FMRP binding assay was repeated in presence of purified recombinant FUS proteins. F1 and F2 biotinylated transcripts were incubated with HeLa extract and purified RFP-flag-FUS^P525L (indicated as P525L) or an RNA-binding deficient mutant derived from RFP-flag-FUS^P525L (indicated as P525L 4F-L). Western blot analysis was performed after pull-down with streptavidin-conjugated beads with anti-FMRP, anti-flag, or anti-GAPDH antibody. Input: 10% of the pull-down input sample. The molecular weight (kDa) is indicated on the left. Histograms show quantification from three independent experiments. Values were calculated as fraction of Input and normalized to P525L (Student's *t*-test; paired; two tails).

HuD protein levels were increased of approximately two-fold (Fig. 3a). In the same cells, HuD transcript levels were unchanged (Fig. 3b), suggesting that absence of FMRP upregulated HuD protein without altering its transcription or mRNA stability. In FMRP^KO MNs we also observed higher transcript and protein levels of two HuD target genes, *NRN1* and *GAP43* (Fig. 3a, b). We next took advantage of a reporter assay to study the outcomes of competitive binding of FMRP and mutant FUS to the *HuD* 3′UTR. We have previously described that expression of a RFP-FUS^P525L transgene led to increased translation of a luciferase construct carrying the *HuD* 3′UTR when compared to RFP-FUS^WT or RFP alone[9]. We repeated the same experiment upon overexpression of FMRP (or eGFP as a control). As shown in Fig. 3c, the *HuD* 3′UTR reporter activity was strongly reduced when FMRP was overexpressed in presence of RFP alone. Notably, co-expression of RFP-FUS^P525L, but not RFP-FUS^WT, partially reverted such negative regulation by FMRP.

Collectively, these results suggest that FMRP can act as negative regulator of HuD translation in MNs by direct 3′UTR binding. Mutant FUS may intrude in this function by competition for 3′UTR binding, resulting in increased HuD protein levels.

**Axon branching and growth phenotypes in FUS mutant MNs.** Since HuD protein levels are two-fold to four-fold higher in human and mouse *FUS* mutant MNs compared to wild-type controls (Fig. 1a, b), we wondered if such upregulation could lead to functional consequences. Overexpression of HuD promotes neurite outgrowth in rat PC12 cells (a cell line of neural crest origin) and cortical neurons and, in vivo, in mice overexpressing HuD[18,19,40]. We therefore took advantage of multichambered microfluidics devices to study possible neurodevelopmental defects in *FUS* mutant MNs. FUS^WT and FUS^P525L human MN progenitors were dissociated and re-plated into one chamber (cell body chamber) of compartmentalized chips with 500 μm microgroove barrier. Such experimental setup allowed us to analyze axons in a separate compartment (axon chamber). Axonal morphology was analyzed after subsequent MN maturation in the device for 7 days. As shown in Fig. 4a, b, we found an increased number of axon branches and branch points in the FUS mutant compared to WT (see also Supplementary Fig. 6a). A similar increase in axon branching was recently reported by Akiyama and colleagues in MNs derived from hiPSCs carrying a different FUS mutation (H517D)[28]. We further extended this analysis by evaluating axon regeneration after damage. At the sixth day of maturation, chemical axotomy was induced by applying trypsin

to the axonal chamber. MNs were then allowed to regenerate their axons for 30 h. Immunostaining of the neuronal tubulin TUBB3 showed strikingly increased outgrowth in FUS^P525L cells (Fig. 4c and Supplementary Fig. 6b, c). Similar results were obtained when axotomy was induced in the same experimental setup by vacuum application (mechanical injury) or digestion with a different chemical agent, accutase (Fig. 4d, e). These results were independently confirmed in the Fus-Δ14 mouse model. Primary embryonic MNs, derived from heterozygous and homozygous mutant mice (E12.5–13.5) and plated in compartmentalized chips, showed increased arborization when compared to WT controls (Fig. 5a, b). Increased re-growth was also observed for axotomized *Fus* mutant mouse MNs (Fig. 5c).

**Increased NRN1 and GAP43 levels in FUS mutant motoneurons.** We next focused on downstream HuD targets, which could be altered as a consequence of increased levels of this RBP in FUS mutant MNs, and that might be involved in the observed axon phenotypes. Among the known HuD targets we prioritized *NRN1* and *GAP43*.

HuD stabilizes the mRNA encoding the growth promoting protein NRN1 (Neuritin1) by binding its 3′UTR[41], and we found *NRN1* mRNA levels to be strongly increased in FUS^P525L MNs as assessed by quantitative RT-PCR (Fig. 6a) and FISH analyses (Fig. 6b). Western blot analysis showed that NRN1 protein levels are negligible in FUS^WT MNs and increased in FUS^P525L cells (Fig. 6c). This observation was confirmed in the mouse model, where P81 spinal cord samples showed increased levels of the mouse Nrn1 homolog in heterozygous and homozygous mutants (Fig. 6d). Immunostaining analysis in human MNs showed that NRN1 is expressed at low levels in the WT axons and is upregulated in the FUS mutant axons (Fig. 6e). Previous work showed that HuD overexpression increased *GAP43* mRNA levels in rat cortical neurons[19]. In particular, HuD binds to an AU-rich regulatory element (ARE) in the 3′UTR of *GAP43* mRNA and stabilizes this transcript[42,43]. As in the case of *NRN1*, increased levels of *GAP43* mRNA were observed in FUS^P525L MNs by quantitative RT-PCR and FISH (Figs. 6a and 7a). An increase of GAP43 protein levels in the FUS mutant background was detected by western blot analysis in hiPSC-derived MNs but not in mouse spinal cord (Fig. 7b, c). The increase in GAP43 protein levels in FUS^P525L human MNs was confirmed by immunostaining analysis. Since HuD binding is known to localize GAP43 at growth cones[44], we focused on these structures and found a striking difference: while GAP43 protein was undetectable in

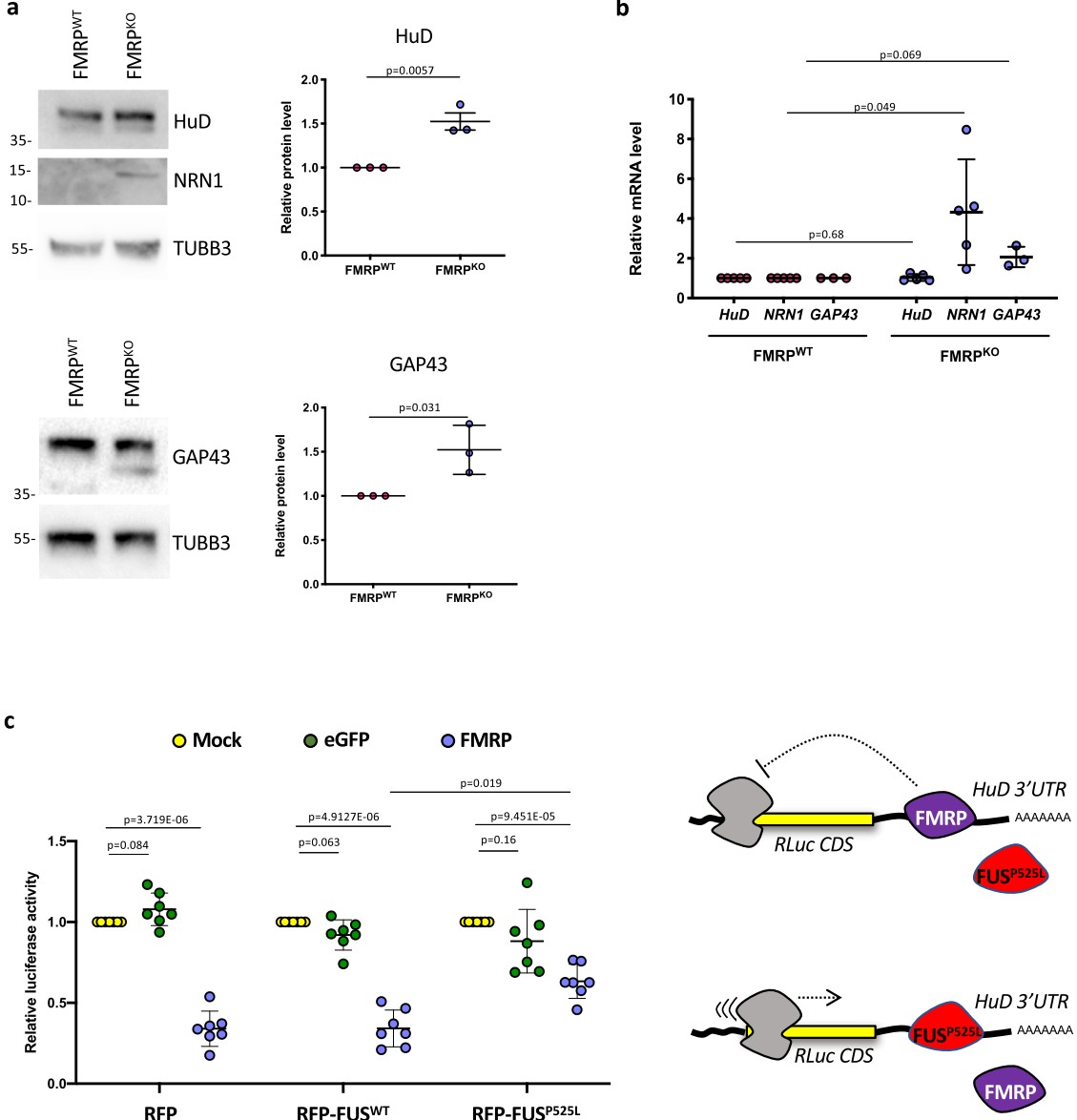

**Fig. 3 FMRP is a post-transcriptional repressor of HuD expression. a** Analysis of the protein levels of the indicated genes by western blot in FMRP[WT] and FMRP[KO] hiPSC-derived spinal MNs. The molecular weight (kDa) is indicated on the left. The graphs show the average from three independent differentiation experiments, error bars indicate the standard deviation (Student's t-test; paired; two tails). TUBB3 signal was used for normalization. Protein levels are relative to FMRP[WT] conditions. **b** Analysis of the mRNA levels of the indicated genes by real-time qRT-PCR in FMRP[WT] and FMRP[KO] hiPSC-derived spinal MNs. For each experiment, values are shown as relative to the isogenic FMRP[WT] control, set to a value of 1. The graph shows the average from 3 to 5 independent differentiation experiments, error bars indicate the standard deviation (Student's t-test; paired; two tails). **c** Luciferase assay in HeLa cells expressing RFP, RFP-FUS[WT], or RFP-FUS[P525L] and transfected with the Renilla luciferase reporter construct containing the *HuD* 3'UTR (RLuc-HuD 3'UTR) alone (Mock) or in combination with plasmids overexpressing FMRP or eGFP as a control (Student's t-test; paired; two tails). The drawing depicts the competition between mutant FUS and FMRP for HuD 3'UTR binding and its effects on the reporter construct.

FUS[WT] MNs, a clear punctate signal was present in the FUS[P525L] mutant (Fig. 7d and Supplementary Fig. 7a). To a minor extent, difference in GAP43 levels was also found in the MN soma (Supplementary Fig. 7b). Increase of *HuD*, *NRN1* and *GAP43* mRNA and protein levels was confirmed in MNs generated from two additional FUS[P525L] hiPSC lines[45] (Supplementary Figs. 8 and 9).

To directly correlate NRN1 and GAP43 upregulation to increased HuD activity, we generated a HuD overexpressing hiPSC line in a FUS[WT] background. Overexpression of HuD in undifferentiated hiPSCs by a constitutive promoter resulted in cell toxicity. We therefore took advantage of a neuronal-specific human synapsin 1 promoter construct (SYN1::HuD) to drive

expression of HuD after induction of MN differentiation (Supplementary Fig. 10a). The SYN1::HuD construct was stably integrated in FUS[WT] hiPSCs (FUS[WT] + HuD in Fig. 6a and in Supplementary Fig. 10b–e). As a control, we also generated a FUS[WT] hiPSC line containing SYN1::RFP construct (FUS[WT] + RFP). These cells were then induced to differentiate to MNs. Rise of HuD in FUS[WT] SYN1::HuD MNs was in the range of HuD levels observed in FUS[P525L] MNs. In these cells, we observed increased levels of *NRN1* and *GAP43* mRNA compared to the parental FUS[WT] line, while no effect was observed in the FUS[WT] + RFP control (Fig. 6a). In the case of *GAP43*, however, change in mRNA levels between FUS[WT] and FUS[WT] + HuD did not reach statistical significance. A stronger effect on *NRN1*,

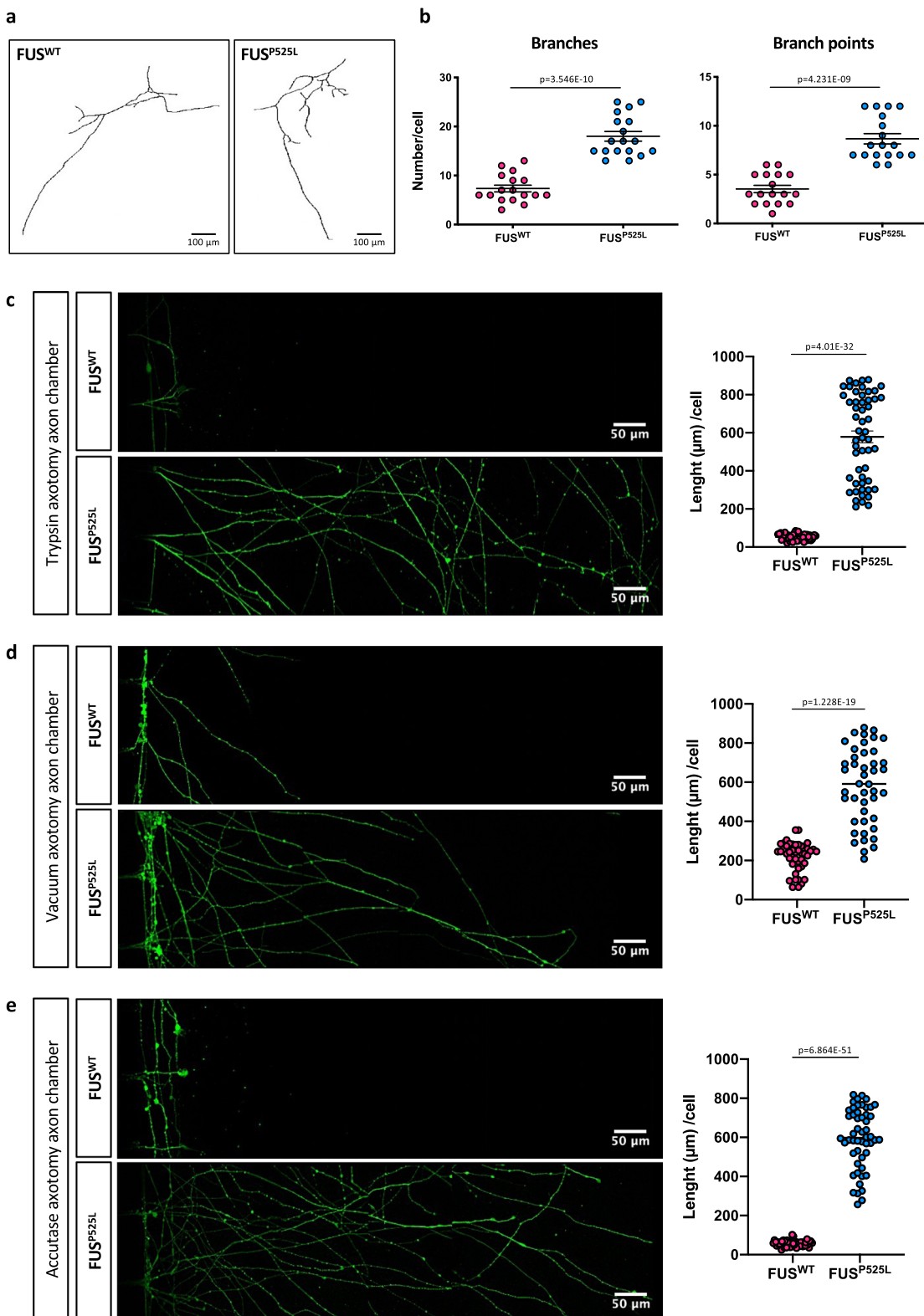

**Fig. 4 Axonal phenotypes in hiPSC-derived spinal MNs. a** Representative images, generated with the Skeleton plugin of ImageJ, showing axons of FUS[WT] and FUS[P525L] hiPSC-derived MNs in the axon chamber of compartmentalized chips. Scale bar: 100 μm. **b** Quantitative analysis of the number of axon branches and branch points in cells shown in **a**. The graphs show the average from three independent differentiation experiments, error bars indicate the standard error of the mean (Student's *t*-test; unpaired; two tails). **c–e** Immunostaining of TUBB3 (green) in FUS[WT] and FUS[P525L] hiPSC-derived spinal MNs cultured in compartmentalized chips and allowed to recover for 30 h after the indicated treatments to induce axotomy in the axon chamber. Scale bar: 50 μm. Graphs show quantitative analysis of axon length from three independent differentiation experiments; error bars indicate the standard error of the mean (Student's *t*-test; unpaired; two tails).

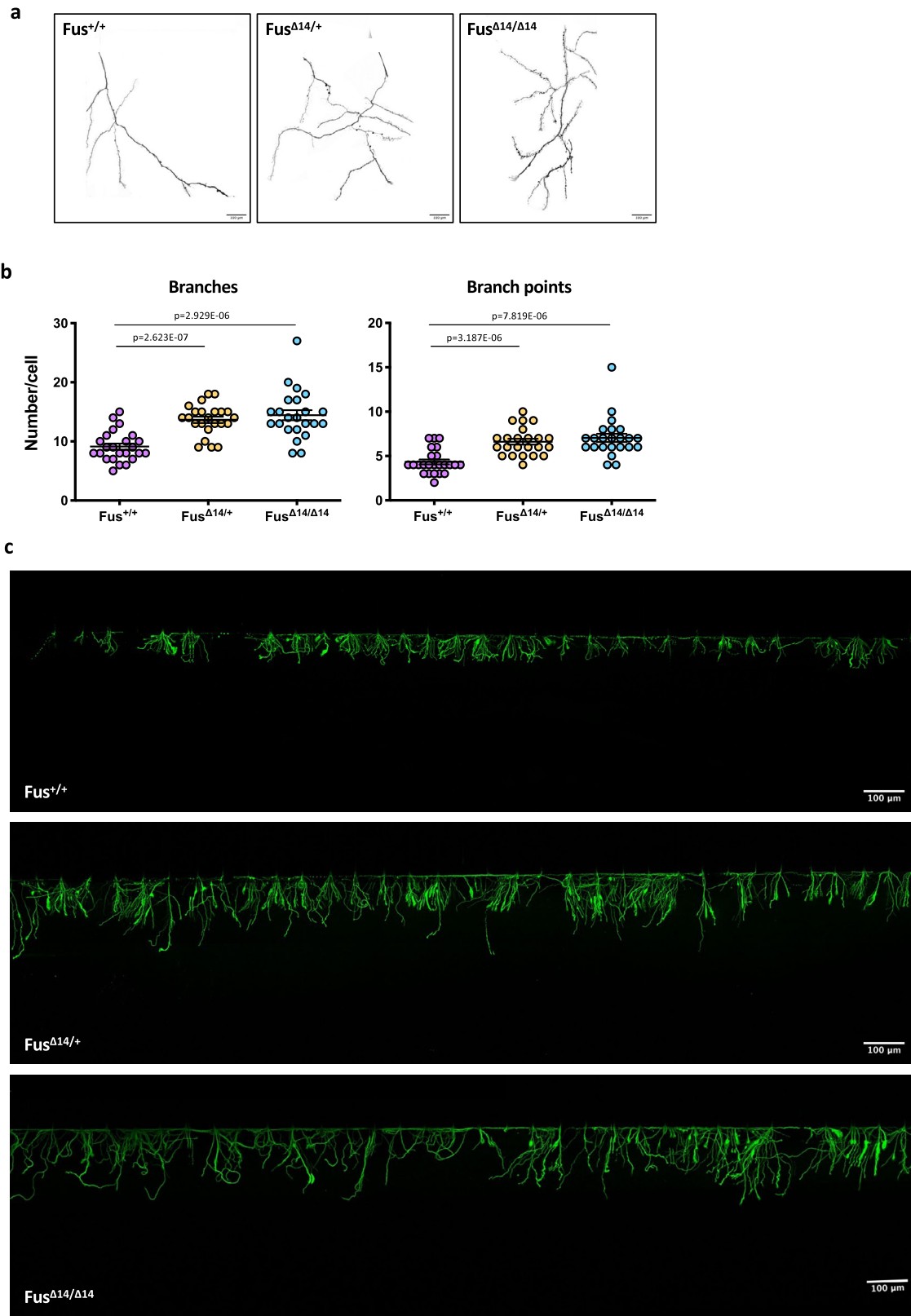

**Fig. 5 Axonal phenotypes in primary spinal MNs from Fus-Δ14 mouse models. a** Representative images, generated with the Skeleton plugin of ImageJ, showing axons of Fus[+/+], and heterozygous (Fus[Δ14/+]) or homozygous (Fus[Δ14/Δ14]) FUS mutant mouse primary MNs in the axon chamber of compartmentalized chips. Scale bar: 100 μm. **b** Quantitative analysis of the number of axon branches and branch points in cells shown in **a**. The graphs show the average from three biological replicates, error bars indicate the standard error of the mean (Ordinary one-way ANOVA; multiple comparisons). **c** Immunostaining of Tubb3 (green) in Fus[+/+], Fus[Δ14/+] and Fus[Δ14/Δ14] mouse primary MNs cultured in compartmentalized chips and allowed to recover for 30 h after trypsin treatment to induce axotomy in the axon chamber. Scale bar: 100 μm.

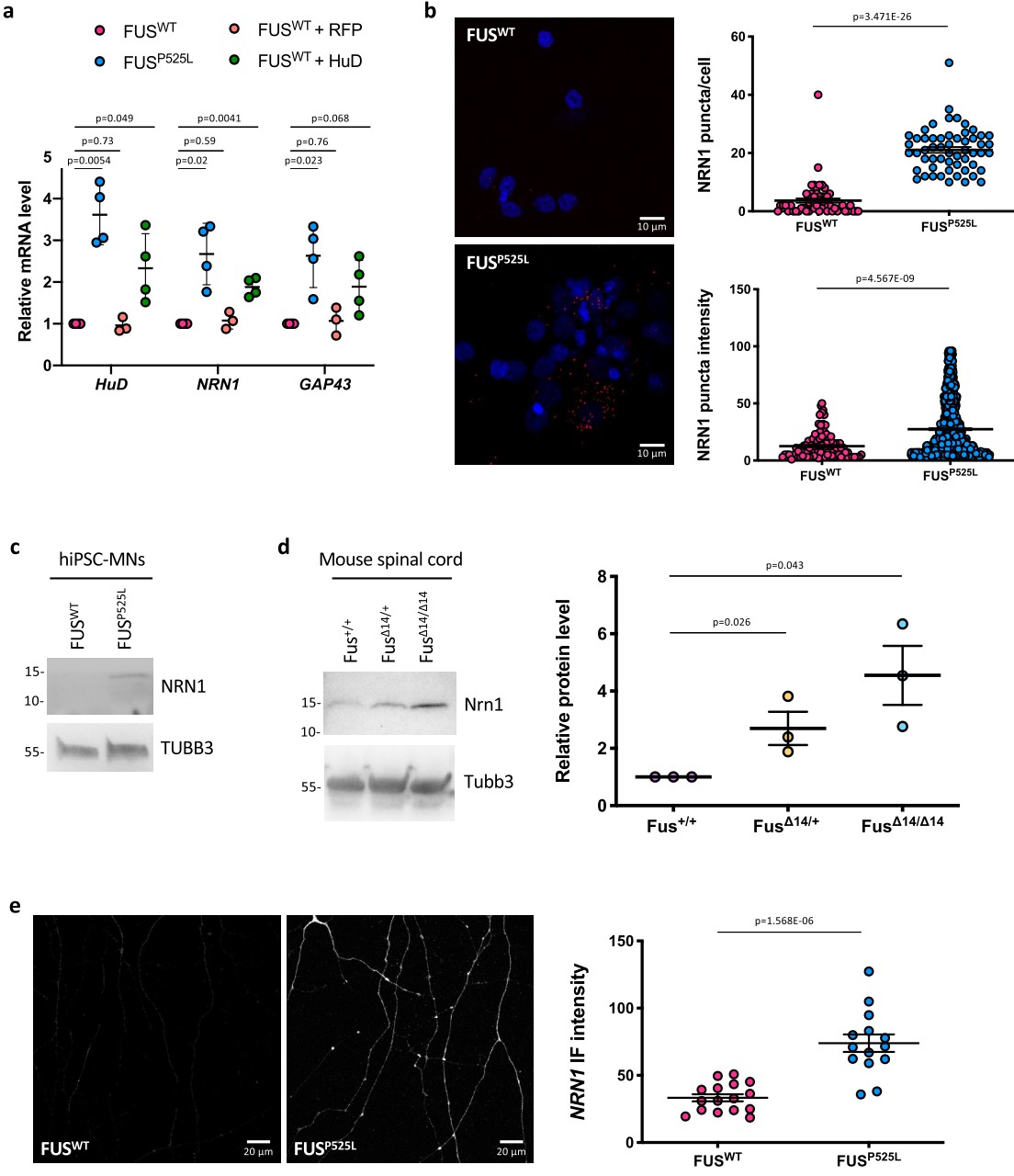

**Fig. 6 NRN1 levels are increased in mutant FUS MNs. a** Analysis of the mRNA levels of the indicated genes by real time qRT-PCR in FUS^WT, FUS^P525L and FUS^WT overexpressing HuD or RFP, as a control, under the SYN1 promoter (FUS^WT + HuD and FUS^WT + RFP) hiPSC-derived spinal MNs. The graph shows the average from three or four independent differentiation experiments, error bars indicate the standard deviation (Student's *t*-test; paired; two tails). **b** NRN1 mRNA analysis by FISH (red) in FUS^WT and FUS^P525L hiPSC-derived spinal MNs. DAPI (blue) was used for nuclear staining. Scale bar: 10 μm. Graphs show the average count of HuD mRNA puncta per cell and the puncta intensity from three independent differentiation experiments, error bars indicate the standard error of the mean (Student's *t*-test; unpaired; two tails). **c, d** NRN1 protein levels analysis by western blot in FUS^WT and FUS^P525L hiPSC-derived spinal MNs (**c**) and Fus-Δ14 mouse model spinal cord (**d**). The molecular weight (kDa) is indicated on the left. The graphs show the average from three independent biological replicates, error bars indicate the standard deviation (ordinary one-way ANOVA; multiple comparisons). Tubb3 signal was used for normalization. Protein levels are relative to Fus^+/+ conditions. **e** Immunostaining analysis of NRN1 in axons of FUS^WT and FUS^P525L hiPSC-derived spinal MNs. The graph shows the NRN1 signal intensity from four replicates from two differentiation experiments, error bars indicate the standard error of the mean (Student's *t*-test; unpaired; two tails).

compared to *GAP43*, was observed also in FMRP^KO MNs, where HuD protein levels were upregulated (Fig. 3a, b).

Collectively these data point to increased levels of HuD targets in FUS mutant MNs as a consequence of the disruption of the FMRP-mediated negative regulation of HuD by FUS^P525L. In particular, NRN1 is strongly upregulated, while GAP43 is affected in the same direction, although to a minor extent.

**Increased axon branching and growth upon axotomy in FUS mutant motoneurons are due to NRN1 upregulation**. Given the greater changes in NRN1 levels, compared to GAP43, in FUS mutant MNs, we decided to prioritize this candidate for further analysis. Increased levels of NRN1 in FUS mutant MNs prompted us to explore the possibility that the phenotypes described in Figs. 4 and 5 are a direct consequence of aberrant activation of

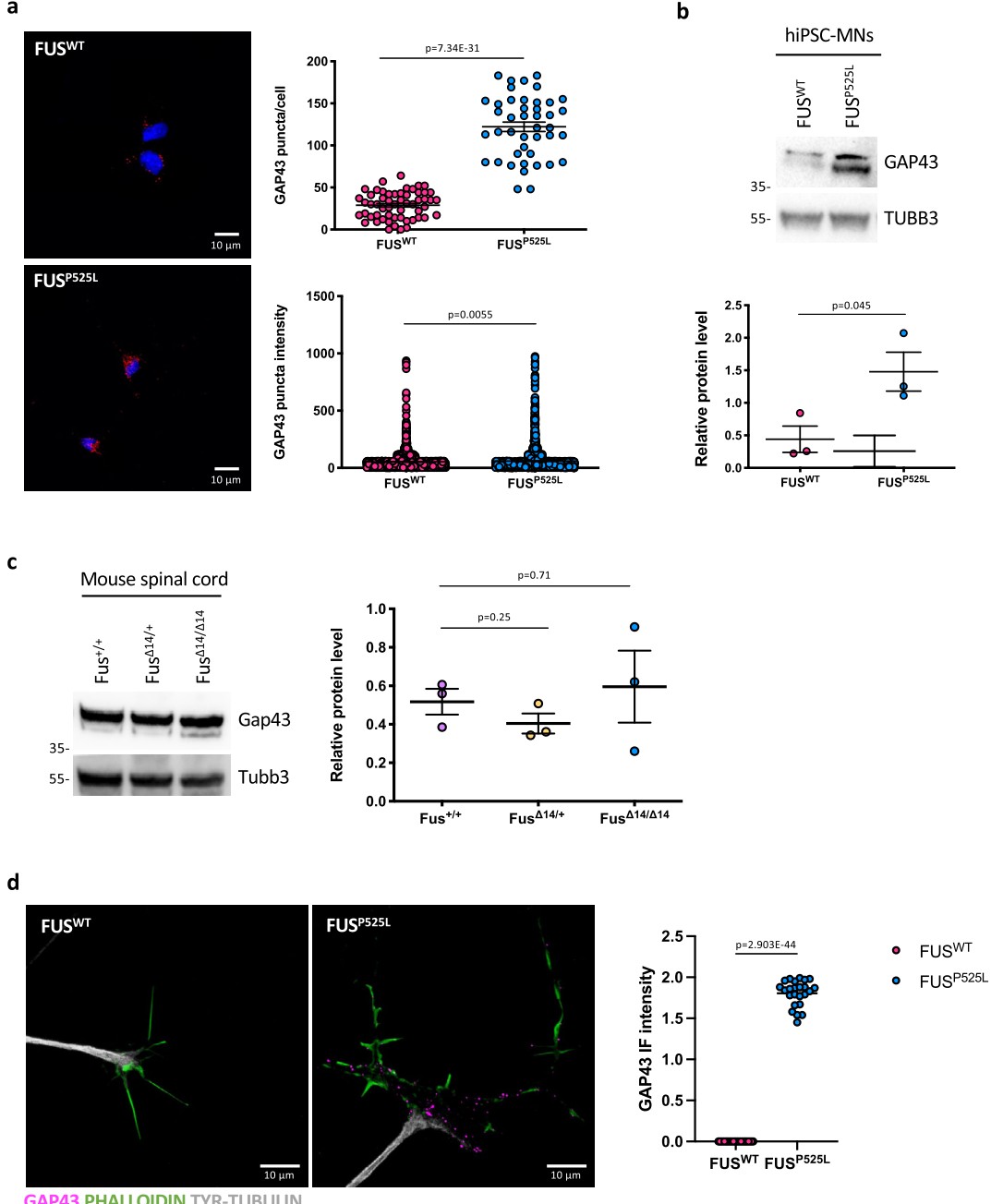

**Fig. 7 GAP43 levels are increased in mutant FUS MNs. a** GAP43 mRNA analysis by FISH (red) in FUS^WT, FUS^P525L, and FUS^WT overexpressing HuD under the Syn1 promoter (FUS^WT + HuD) hiPSC-derived spinal MNs. The graphs show the average count of HuD mRNA puncta per cell from three independent differentiation experiments, error bars indicate the standard error of the mean (Student's *t*-test; unpaired; two tails). DAPI (blue) was used for nuclear staining. Scale bar: 10 μm. **b** GAP43 protein levels analysis by western blot in FUS^WT and FUS^P525L hiPSC-derived spinal MNs. The molecular weight (kDa) is indicated on the left. The graph shows the average from three independent differentiation experiments and error bars indicate the standard deviation (Student's *t*-test; paired; two tails). Values in the *y*-axis represent GAP43/TUBB3 signal intensity. **c** Gap43 protein level analysis by western blot in mouse primary spinal MNs (P81). The molecular weight (kDa) is indicated on the left. The graphs show the average from three mice and error bars indicate the standard deviation. The differences are not significant for all pairs (ordinary one-way ANOVA; multiple comparisons). Values in the *y*-axis represent Gap43/Tubb3 signal intensity. **d** Immunostaining analysis in FUS^WT and FUS^P525L hiPSC-derived spinal MNs growth cones. GAP43 signal is magenta; PHALLOIDIN signal (marking growth cones) is green, TYR-TUBULIN (tyrosinated alpha-tubulin; marking axons) is white. Scale bar: 10 μm. The graph shows the GAP43 signal intensity from three differentiation experiments, error bars indicate the standard error of the mean (Student's *t*-test; paired; two tails).

this growth promoting protein. We addressed this hypothesis by a rescue approach. Small interfering RNAs (siRNAs), transfected during hiPSC-derived MN maturation, effectively reduced NRN1 levels in the FUS^P525L background (Fig. 8a). In siRNA-NRN1-treated FUS^P525L MNs cultured in microfluidics devices we

observed reduced number of axon branches and branch points, compared to non-targeting control siRNAs (Fig. 8b, c). We next performed in these cells the trypsin-induced axotomy and regeneration assay as in Fig. 4c. TUBB3 immunostaining analysis showed that axon growth after regeneration was strongly reduced

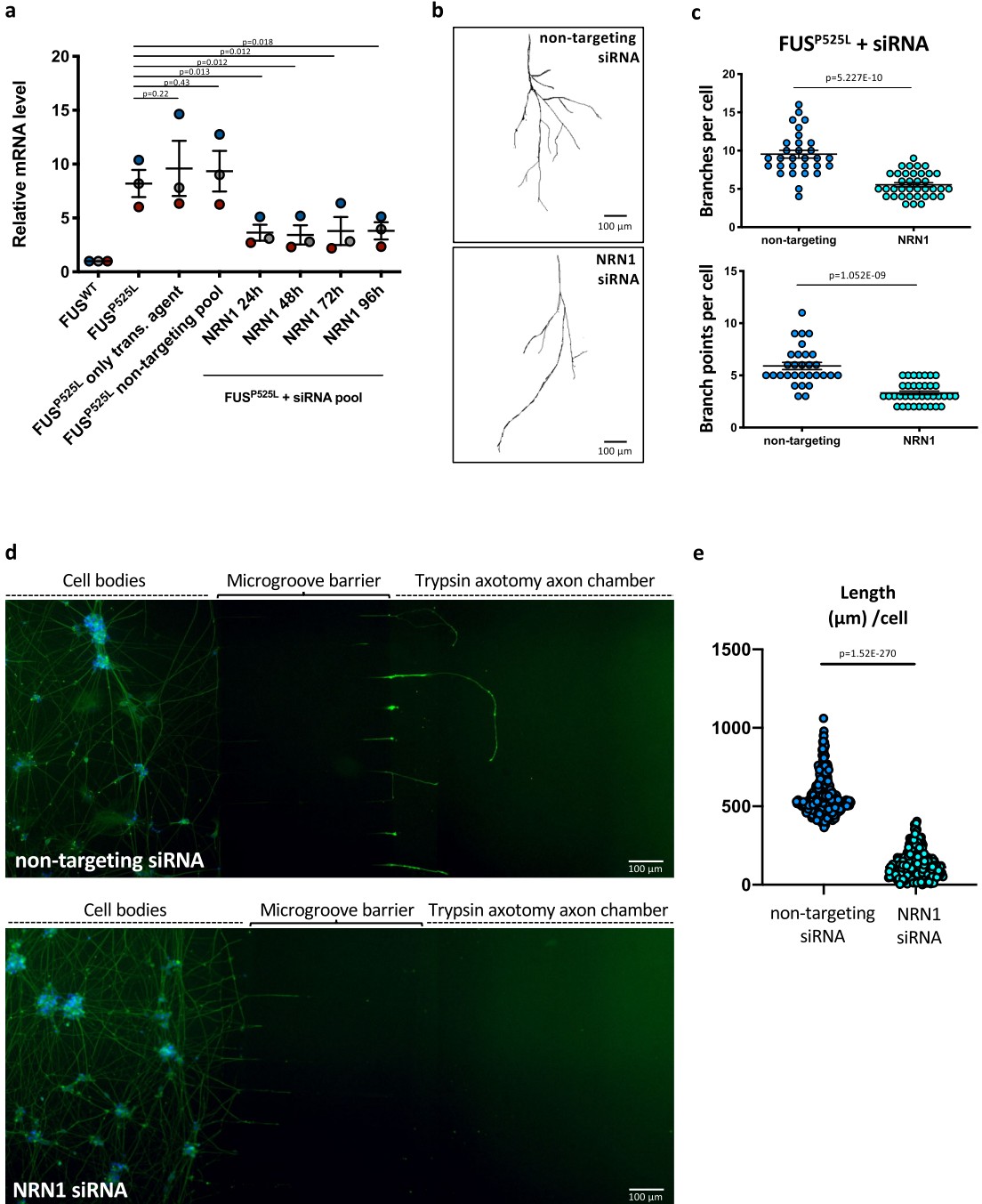

**Fig. 8 NRN1 knockdown rescues the aberrant axon growth phenotype in mutant FUS MNs. a** Analysis of the mRNA levels of the indicated genes by real time qRT-PCR in untransfected FUS[WT] and FUS[P525L] hiPSC-derived spinal MNs and FUS[P525L] hiPSC-derived spinal MNs transfected with non-targeting control siRNAs or siRNAs targeting NRN1. Values are shown as relative to untransfected FUS[WT] samples. The graph shows the average from three independent transfection experiments (indicated by different color of the dots), error bars indicate the standard deviation (Student's *t*-test; paired; two tails). **b** Representative images, generated with the Skeleton plugin of ImageJ, showing axons of FUS[P525L] hiPSC-derived MNs, transfected with the indicated siRNA pools, in the axon chamber of compartmentalized chips. Scale bar: 100 μm. **c** Quantitative analysis of the number of axon branches and branch points. Immunostaining of TUBB3 was carried out 5 days after transfection of the indicated siRNA pools in FUS[P525L] hiPSC-derived MNs cultured in compartmentalized chips. The graphs show the average from five independent transfections of non-targeting or NRN1 siRNAs from three differentiation experiments, error bars indicate the standard error of the mean (Student's *t*-test; unpaired; two tails). **d** Immunostaining of TUBB3 (green) in FUS[P525L] hiPSC-derived spinal MNs cultured in compartmentalized chips, transfected with non-targeting control siRNAs or siRNAs targeting NRN1, treated with trypsin in the axon chamber to induce axotomy after 24 h, and allowed to recover for 24 h. DAPI (blue) was used for nuclear staining. Scale bar: 100 μm. **e** Graph showing quantitative analysis of axon length in MNs treated as in **d** from seven independent transfections of non-targeting or NRN1 siRNAs from two differentiation experiments, error bars indicate the standard error of the mean (Student's *t*-test; unpaired; two tails).

in FUS[P525L] MNs treated with NRN1-siRNAs (Fig. 8d, e). These results suggest that increased axon branching and growth observed in FUS mutant MNs are mediated by higher levels of NRN1 and that knock-down of NRN1 is sufficient to revert these phenotypes.

## Discussion

Here, we propose a regulatory mechanism for *HuD* translation in normal MNs and its increase in ALS. A relevant consequence of HuD upregulation in FUS mutant MNs is the increase of two HuD targets: NRN1 and GAP43. In turn, NRN1 hyperactivation confers aberrantly increased axon branching and growth upon axotomy to FUS mutant MNs.

According to our model (Fig. 9), a mutation that impairs the nuclear localization of FUS may trigger a domino effect onto other RBPs. One of the consequences is the escape of *HuD* from negative regulation by FMRP on its 3′UTR. The highly conserved *HuD* 3′UTR is indeed a relevant regulatory element, with an important role in keeping HuD protein levels in check. For this purpose, we propose that at least two distinct mechanisms are in place in MNs. The first one involves the activity of the MN-enriched microRNA, miR-375[17]. The second one, described in this work, relies on the negative regulation of translation by FMRP. Notably, both mechanisms are impaired by FUS mutations. Since *HuD* mRNA levels are affected by miR-375[9] but not by FMRP (present work), we can conclude that in mutant FUS MNs increased *HuD/ELAVL4* mRNA levels are due to decreased miR-375 expression[17], while increased HuD/ELAVL4 protein levels can be due to the double effect of the loss of both miR-375 and of FMRP regulation. Interestingly, impaired miR-375 function has been also proposed by others in a mouse model of sporadic ALS[46] and in another MN disease, type1 SMA[47]. In addition to miR-375 and FMRP, our *cat*RAPID analysis indicates that the *HuD* 3′UTR might be also a target of the HuD protein (Supplementary Table 1), thus suggesting possible conservation in human of the autoregulatory mechanism previously proposed in Drosophila and mouse[48,49].

Impairment of FMRP-mediated repression of *HuD* in ALS FUS mutant MNs might occur in several, non-exclusive, possible ways. First, FMRP might be captured in mutant FUS insoluble aggregates, as proposed by Blokhuis and colleagues[14], who also reported impaired FMRP-mediated translational repression and altered MAP1B protein levels in cells overexpressing mutant FUS. In both FUS[P525L] hiPSC-derived MNs and Fus-Δ14 mice, however, mutant FUS is expressed at physiological levels and does not form aggregates[9,33,50]. Second, FUS mutations might promote phase separation of FMRP by sequestering it in FUS-containing cytoplasmic ribonucleoprotein complexes (RNPs)[51]. Third, mutant FUS might directly compete with FMRP for 3′UTR binding. HuD 3′UTR contains multiple putative regulatory elements and competitive or cooperative 3′UTR binding is a regulatory mechanism extensively used by RBPs[52,53]. Interestingly, loss of the FMRP homolog dFXR leads to NMJ defects in Drosophila[54], while exogenous FMRP expression rescued NMJ and locomotor defects in a zebrafish FUS ALS model[14]. Recent evidence has also linked FMRP with TDP-43[55,56], suggesting that FMRP involvement in ALS might extend beyond FUS.

Increased HuD suggested a possible underlying mechanism for the increased axon branching and growth phenotypes that we observed in both human and mouse ALS FUS models. HuD has indeed a well-known role in promoting neurogenesis in cell lines and cortical neurons[18,19]. However, to our knowledge, the role of HuD in MNs has never been specifically addressed. Here, we show that at least two relevant HuD targets are upregulated in FUS mutant MNs as a consequence of loss of HuD repression:

GAP43 and NRN1. GAP43 is upregulated downstream of increased HuD during axon regeneration upon sciatic nerve injury[44]. In a transgenic mouse model, overexpression of GAP43 induces prolonged nerve sprouting and causes death of adult MNs[57,58]. Together with our present findings, those observations suggest that GAP43 aberrant increase might in part contribute to the pathogenic effects of FUS mutations in ALS MNs.

The growth promoting protein NRN1 is one of the primary HuD targets in MNs. HuD stabilizes NRN1 mRNA via AU-rich element (ARE) binding on its 3′UTR[41,59]. Overall, we observed stronger effects on NRN1 compared to GAP43, at both mRNA and protein levels (with the relevant exception of GAP43 at the growth cone). This is in agreement with previous findings showing that the NRN1 ARE has a higher binding affinity for HuD compared to GAP43 ARE[60]. NRN1, also known as CPG15, was first identified as a candidate plasticity-related gene (CPG) induced by the glutamate analog kainate in the hippocampus dentate gyrus, along with immediate early genes (IEGs) such as c-Fos and c-Jun[61]. It was later demonstrated that NRN1 is an activity-regulated IEG induced by calcium influx through NMDA receptors and L-type voltage-sensitive calcium channels[62]. Its expression in the rat neocortex peaks at 14 days postnatal and then decreases in the adult[63]. In the adult rat, NRN1 mRNA is detected in brain regions characterized for their activity-modulated plasticity (hippocampus, olfactory bulb, and Purkinje cells), and can be induced by glutamate analogs, neurotrophins (such as BDNF) and neural activity[64]. In the Human Protein Atlas[65], the spinal cord is reported among the nervous system regions with lowest NRN1 expression (Supplementary Fig. 11). We found increased NRN1 levels in FUS mutant hiPSC-derived MNs and mouse spinal cord, in the absence of promoting stimuli. NRN1 was also upregulated upon HuD overexpression and FMRP knock-out. When overexpressed in rodent or Xenopus neurons, NRN1 induced neurite outgrowth, elaboration of dendritic and axonal arbors and synaptic maturation by AMPA receptor insertion[64,66,67]. Moreover, axonal localization of *Nrn1* mRNA, which is induced after nerve injury and is mediated by the 3′UTR in central nervous system and by the 5′UTR in peripheral nervous system axons, promotes axon growth[68]. NRN1 is highly expressed in developing MNs, where its overexpression increases axonal outgrowth and neurite branching[69]. This phenotype is remarkably similar to increased neurite outgrowth and branching observed in motor neurons expressing the ALS SOD1-G93A mutant[29]. Consistently, increased axon branching and growth upon axotomy occurring in our mutant FUS models could be rescued by reduction of NRN1 levels with siRNAs. This result suggests that increased NRN1 mRNA stability, downstream of HuD upregulation, may be one of the key aberrant mechanisms underlying the observed axon phenotypes.

Consistent with our findings, Akiyama and colleagues have recently reported an increased axon branching phenotype in hiPSC-derived MNs carrying the FUS[H517D] mutation[28]. They also showed increased levels of AP-1 components (including members of the FOS family) and reversion of the axon branching phenotype upon FOS-B reduction[28]. AP-1 increase might not directly contribute to NRN1 upregulation in FUS mutant cells. In stimulated cortical neurons, indeed, NRN1 upregulation occurred independently from new protein synthesis, indicating that NRN1 induction does not require prior activation of AP-1[62]. Therefore, the effects observed by Akiyama and colleagues upon modulation of FOS-B are unlikely mediated by NRN1. To the best of our knowledge, NRN1 has been never associated with ALS before. However, by mining published RNA-Seq datasets we found an upregulation trend of the *Nrn1/NRN1* transcript in the soma of MNs dissected from a SOD1 mouse model at a pre-symptomatic stage (3 months)[70] or generated from SOD1 mutant human and mouse stem cells[71]. Pro-

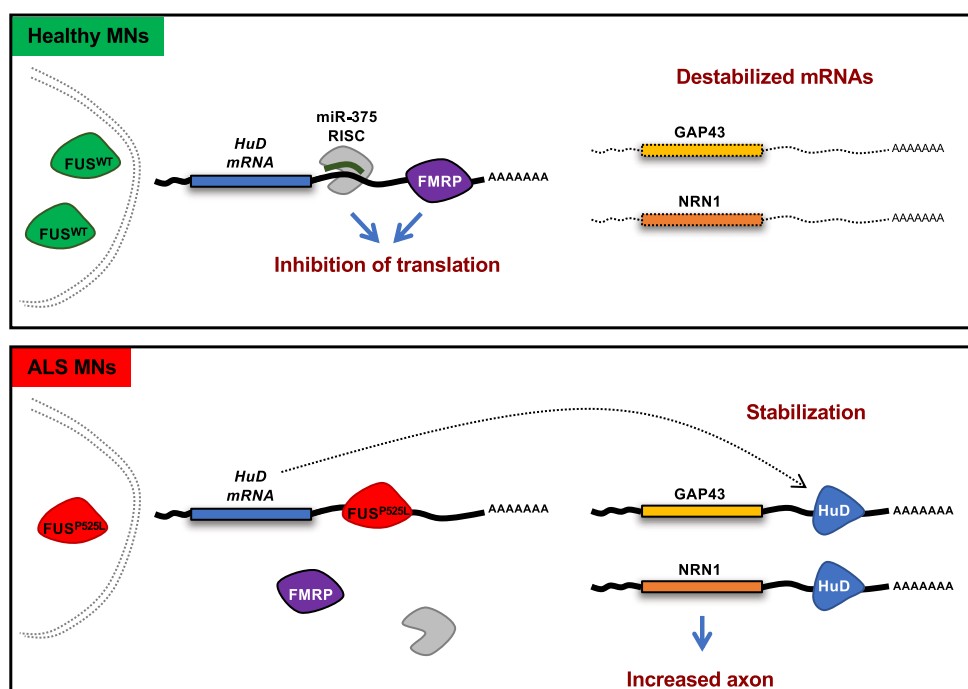

**Fig. 9 Model depicting the proposed molecular mechanism underlying HuD regulation by FMRP and mutant FUS.** The figure depicts a model of the competition between mutant FUS and FMRP for HuD 3'UTR binding. In FUS[WT] MNs, the FUS protein is predominantly localized in the nucleus. In the cytoplasm, FMRP binds HuD 3'UTR repressing its translation. NRN1 mRNAs are destabilized. In FUS[P525L] MNs, mutant FUS is partially delocalized to the cytoplasm and outcompetes FMRP binding on the HuD 3'UTR. As a consequence, increased HuD protein levels accumulate in FUS mutant MNs. HuD binding to NRN1 and GAP43 3'UTR leads to stabilization of these transcripts and higher protein levels. NRN1 increase underlies the aberrant axonal growth phenotypes.

regenerative effects on adult MNs upon ALS-mutant SOD1 expression have been recently shown, including increased out-growth and branching[29]. This is unlikely a compensatory response to mutant SOD1-induced toxicity, as it occurs also upon acute expression of mutant SOD1 in WT adult MNs. Notably, MN axon branching was also increased in zebrafish embryos injected with ALS-mutant TDP-43 proteins, which showed a motor deficit[26]. The importance of aberrant axon branching and growth in ALS pathophysiology deserves more investigation, as it might be detrimental for the normal function of signal transmission in MNs[24]. Notably, recent evidence of HuD upregulation and increased binding activity in sporadic ALS patients' motor cortex[72] suggests that the present findings might extend beyond FUS-ALS. Our work provides insights into the molecular mechanisms underlying such axonal phenotypes in FUS-ALS.

## Methods

**Plasmids construction.** The epB-Puro-TT-SYN1-HuD (SYN1::HuD) and epB-Puro-TT-SYN1-RFP (SYN1::RFP) plasmids were generated by inserting the sequences of the human synapsin 1 (SYN1) promoter and, respectively, Flag-HA-HuD and tagRFP in the enhanced piggyBac transposable vector epB-Puro-TT[73]. The SYN1 promoter was isolated from the eMSCL WT plasmid (Addgene, #107454) via PCR with the following primers: hSYN1 FW 5′-CATCTCGAGCAG TGCAAGTGGGTTTTAGGAC-3′; hSYN1 RV 5′-CATGGATCCACTGCGCTCT CAGGCACGA-3′. Flag-HA-HuD was obtained by cutting the pFRT-TODestFLA GHA_HuD plasmid (Addgene, #65757) with PaeI and BglII enzymes. Such HuD sequence is devoid of both 5′ and 3′UTR. The resulting constructs contain the enhanced piggyBac terminal repeats flanking a constitutive cassette driving the expression of the puromycin resistance gene fused to the rtTA gene and, in the opposite direction, a SYN1 promoter driving the expression of the transgenes (Supplementary Fig. 10). The epB-Bsd-TT-FMR1 and epB-Bsd-TT-eGFP plasmids were generated by subcloning the FMR1 and eGFP coding sequences, respectively, in the enhanced piggyBac transposable vector epB-Bsd-TT[9].

**hiPSC culture, differentiation, and transfection.** Human iPSC lines used in this study were previously generated and characterized: FUS[WT] and FUS[P525L] (WT I and FUS-P525L/P525L lines[50], respectively), KOLF WT 2 and P525L16 (LL FUS-eGFP), and T12.9 WT15 and P525L17 (SL FUS-eGFP) (a kind gift of J. Sterneckert[45]). As indicated in the original studies, informed consent had been obtained from all patients involved prior to cell donation. Cells were maintained in Nutristem XF/FF (Biological Industries) in plates coated with hESC-qualified matrigel (Corning) and passaged every 4–5 days with 1 mg/ml dispase (Gibco). The FMR1 KO hiPSC line[36] was generated by CRISPR/Cas9 gene editing from the WT I line as described in Supplementary Fig. 4. HiPSCs were co-transfected with 4.5 μg of transposable vector (epB-NIL, SYN1::HuD, SYN1::RFP) and 0.5 μg of the piggyBac transposase with the Neon Transfection System (Life Technologies), using 100 μl tips in R buffer and the settings: 1200 V, 30 ms, 1 pulse. Selection was carried out in 5 μg/ml blasticidin S (for epB-NIL) and 1 μg/ml puromycin (for SYN1::HuD, SYN1::RFP), giving rise to stable cell lines.

To obtain spinal MNs, hiPSCs were stably transfected with epB-NIL, an inducible expression vector containing the Ngn2, Isl1, and Lhx3 transgenes[31,32], and differentiated by induction with 1 μg/ml doxycycline (Thermo Fisher Scientific) in DMEM/F12 (Sigma-Aldrich), supplemented with 1× Glutamax (Thermo Fisher Scientific), 1× NEAA (Thermo Fisher Scientific) and 0.5× Penicillin/Streptomycin (Sigma-Aldrich) for 2 days and Neurobasal/B27 medium (Neurobasal Medium, Thermo Fisher Scientific; supplemented with 1× B27, Thermo Fisher Scientific; 1× Glutamax, Thermo Fisher Scientific; 1× NEAA, Thermo Fisher Scientific; and 0.5× Penicillin/Streptomycin, Sigma Aldrich), containing 5 μM DAPT and 4 μM SU5402 (both from Sigma-Aldrich) for additional 3 days. At day 5, MN progenitors were dissociated with Accutase (Thermo Fisher Scientific) and plated on Matrigel (BD Biosciences)-coated 15 mm diameter dishes or cover glass (0.13–0.17 thick), or bipartite/tripartite microfluidic chambers (MFCs, see below) at the density of 10^5 cells per cm^2. Ten micromolar of rock inhibitor was added for the first 24 h after dissociation. Neuronal cultures were maintained in neuronal medium (Neurobasal/B27 medium supplemented with 20 ng/ml BDNF and 10 ng/ml GDNF, both from PreproTech; and 20 ng/ml L-ascorbic acid, Sigma-Aldrich). MFCs were made with Sylgard 184 silicone elastomer kit (Dow Corning) using epoxy resin molds. Once the MFCs were baked, reservoirs were cut and the MFCs were mounted onto glass-bottom dishes (HBST-5040, WillCo well), pre-coated with 1:200 100× poly-D-Lysine. MFCs were then blocked with 0.8% BSA in ES (Sigma-Aldrich) overnight and then coated with Matrigel (BD Biosciences), before plating MN progenitors. MFCs have 500 μm long grooves that separate the somatic from the axonal compartment.

**Mouse primary motoneurons**. Mouse primary motoneurons were obtained from the Fus-Δ14 model (B6N;B6J-Fus^tm1Emcf/H, MGI: 6100933)[33]. All applicable international, national, and institutional guidelines, including ARRIVE guidelines, for the care and use of animals were followed. All procedures performed in studies involving animals were in accordance with the ethical standards of the institution at which the studies were conducted (University College London, UK; MRC Harwell Institute, Oxfordshire, UK). All procedures for the care and treatment of animals were in accordance with the Animals (Scientific Procedures) Act 1986 Amendment Regulations 2012. Primary MNs (PMNs) were isolated from E12.5–13.5 mouse embryos on a congenic C57BL/6J background. Briefly, embryos were euthanized, spinal cord removed and ventral regions isolated. PMNs were dissociated by incubation with trypsin, followed by mechanical dissociation in combination with DNase treatment. Cells were then centrifuged through a bovine serum albumin (BSA) cushion and resuspended in motor neuron medium (Neurobasal; Thermo Fisher, Waltham, MA), 2% v/v B27 supplement (Thermo Fisher Scientific), 2% heat-inactivated horse serum (HRS), 1× GlutaMAX (Thermo Fisher Scientific), 24.8 μM β-mercaptoethanol, 10 ng/ml rat ciliary neurotrophic factor (CNTF; R&D Systems), 0.1 ng/ml rat glial cell line-derived neurotrophic factor (GDNF; R&D systems), 1 ng/ml human brain-derived neurotrophic factor (BDNF; PeproTech) and 1% penicillin/streptomycin. PMNs were immediately plated on poly-L-ornithine/laminin-coated plates or MFCs and cultured for 6–7 days at 37 °C in a 5% CO$_2$ incubator.

Adult spinal cord samples were collected from female Fus-Δ14 mice on a (C57BL/6J × DBA/2J) F1 hybrid background, via laminectomy, and snap frozen over liquid nitrogen. The hybrid background was necessary to produce viable homozygotes, which are non-viable on a congenic C57BL/6J background.

**Western blot**. Western blot analysis was carried out using anti-HuD (1:1000; sc-48421, Santa Cruz), anti-NEURITIN (1:200; AF283, R&D Systems), anti-GAP43 (1:500; LS-C356053, Bio-techne) (for human samples), anti-GAP-43 (1:500; 5307, Cell Signaling Technology) (for mouse samples), anti-flag (1:1000; F3165, Sigma-Aldrich), anti-TUBB3 (1:10000; T2200, Sigma-Aldrich), anti-GAPDH (1:2000; MAB-10578 Immunological sciences) primary antibodies and donkey anti-mouse IgG (H+L) (IS20404; Immunological Science) and donkey anti-rabbit IgG (H+L) (IS20405; Immunological Science) secondary antibodies, with NuPAGE 4–12% Bis–Tris gels (Life Technologies) in MOPS-SDS buffer (Thermo Fisher Scientific). Signal detection was performed with the Clarity Western ECL kit (Bio-Rad). Images acquisition was performed with a Chemidoc MP (Bio-Rad), using the ImageLab software (Bio-Rad) for protein levels quantification. Uncropped blots are shown in Supplementary Figs. 12 and 13.

**Real-time qRT-PCR**. Total RNA, extracted with the RNA extract kit (1 × 10^6 cells-10 mg; VWR International PBI) and retrotranscribed with iScript Supermix (Bio-Rad Laboratories), was analyzed by real-time qRT-PCR with iTaq Universal SYBR Green Supermix (Bio-Rad Laboratories). *ATP5O* was used as the internal calibrator. Primers sequences are listed in Supplementary Table 2.

**RNA interference (RNAi)**. NRN1 RNAi was induced with a synthetic siRNA pool targeting the human NRN1 mRNA (NRN1 51299 siRNA-SMARTpool, Dharmacon) or with a non-targeting siRNA control pool with scrambled sequence (ON-TARGETplus Non-targeting Pool; D-001810-10-05, Dharmacon). Lyophilized siRNAs were resuspended in nuclease-free water and stored at −20 °C as 20 μM stocks until use. For transfection, siRNAs were diluted in Optimem (Gibco) and mixed with siLentFect Lipid Reagent (Bio-Rad Laboratories), according to the protocol for transfection of adherent cells. The final siRNA concentration in each MFC was 10 nM. Medium was changed 5 h of post-transfection.

**Immunofluorescence**. Cells were fixed in 4% paraformaldehyde (PFA) in PBS for 10 min at room temperature. Coverslips were washed with PBS, permeabilized and blocked for 15 min using a solution of 0.5% BSA, 10% HRS, 0.2% Triton X-100 in PBS (all from Sigma-Aldrich). Anti-TUJ1 (for TUBB3 detection; 1:1000; T2200, Sigma-Aldrich), anti-GAP-43 (1:500; LS-C356053, Bio-techne), anti-MAP2 (1:2000; ab5392, Abcam), anti-NEURITIN (1:200; AF283, R&D Systems), anti-488-PHALLOIDIN (1:50; 49429, Sigma-Aldrich), Anti-TYR-TUBULIN clone YL1/2 (1:1000; MAB1864-I, Sigma-Aldrich) antibodies were diluted in a solution of 0.5% BSA, 10% HRS in PBS and incubated with the cells for 1 h at room temperature. Cells were then washed in PBS and incubated for 1 h at room temperature with the appropriate fluorescent conjugated secondary antibodies: anti-mouse Alexa Fluor 488 (1:200, Thermo Fisher Scientific), anti-rabbit Alexa Fluor 594 (1:200, Immunological Sciences) and anti-goat Alexa Fluor 594 (1:200, Immunological Sciences) produced in donkey; anti-Rat IgG (H+L), highly cross-adsorbed, CF™ 647 secondary antibody (1:500; SAB4600186, Sigma-Aldrich) produced in goat; DAPI (1:2000; Sigma-Aldrich) diluted in 0.5% BSA, 10% HRS in PBS. Finally, cells were washed with PBS, mounted using DAKO mounting media and imaged using inverted Zeiss LSM 780 or 510 confocal microscopes using a 63×, 1.4 NA DIC Plan-Apochromat oil-immersion objective, except for images shown in Fig. 8d, which have been acquired with an inverted Olympus iX73 equipped with an X-Light V3 spinning disc head (Crest Optics), a LDI laser illuminator using 470 nm

wavelength (89 North), a CoolSNAP MYO CCD camera (Photometrics) and MetaMorph software (Molecular Devices) with a 10× objective.

**Puro-PLA**. Cells were treated with either DMSO or 40 μM anisomycin for 30 min. Then, 2 μM puromycin was added to the medium for 7 min at 37 °C in a 5% CO$_2$ incubator, cells were washed two times with PBS and fixed in 4% PFA in PBS for 12 min at room temperature. Coverslips were permeabilized with 0.2% Triton X-100 in PBS for 15 min and blocked using a solution of 0.5% BSA and 10% HRS in PBS for 30 min. Detection of newly synthesized proteins was carried out by an anti-puromycin antibody (clone 12D10, mouse-monoclonal, MABE343; Merck), an anti-HuD antibody (ab96474, rabbit-monoclonal; Abcam) and Duolink PLA Fluorescence reagents Red (DUO92008, Duolink), according to manufacturer's instructions, using rabbit PLAplus (DUO92002, Duolink) and mouse PLAminus (DUO92004, Duolink). The anti-TUJ1 antibody was used as a cell marker and to identify axons, as described in the "Immunofluorescence" section. All samples were mounted in Duolink In Situ Mounting Media with DAPI (DUO82040, Duolink).

**Fluorescence in situ hybridization (FISH)**. Cells were fixed for 15 min with 4% PFA in PBS. FISH was performed with the QuantiGene ViewRNA ISH Cell Assay (Thermo Fisher Scientific) protocol for adherent cells. Briefly, fixed MNs were rinsed three times in PBS containing 5 mM MgCl$_2$ for 5 min. Cells were then dehydrated in ethanol (50, 75, and 100%) for 2 min each and stored at −20 °C for up to one week, minimum of 2 hours. Coverslips were rehydrated (75%, 50%) for 2 min each and washed three times in PBS containing 5 mM MgCl$_2$ for 5 min. Coverslips were permeabilized with Detergent Solution and then treated with Protease QS (1:8000). FISH was carried out following manufacturer's instructions (QVC000; Thermo Fisher Scientific). Cells were mounted using DAKO mounting media or Prolong Gold with DAPI (Invitrogen) and imaged using an inverted Zeiss LSM 780 or 510 confocal microscope using a 63×, 1.4 NA DIC Plan-Apochromat oil-immersion objective and Zeiss LSM 880 laser scanning confocal microscope.

**RNA immunoprecipitation (RIP)**. HiPSC-derived motor neurons at day 12 of differentiation were lysed with PLB Buffer (5 mM MgCl$_2$, 10 mM HEPES (pH 7.0), 150 mM KCl, 5 mM EDTA (pH 8), 0.5% NP-40, 2 mM DTT, with 100 U/ml RNAse inhibitor and 1× protease inhibitor cocktail), incubated for 5 min on ice and centrifuged 10 min at 4 °C at 14,000×g. Protein concentration of the supernatant was then measured by Bradford assay and a volume containing 1 mg of proteins was diluted in NT2 Buffer (50 mM Tris (pH 7), 150 mM NaCl, 0.5 mM MgCl$_2$, 0.05% NP-40, 1 mM DTT, 20 mM EDTA (pH 8) with 100 U/ml RNAse inhibitor and 1× protease inhibitor cocktail). Protein G-coupled dynabeads (immunoprecipitation kit, Invitrogen) were washed in NT2 Buffer, incubated with 10 μg of anti-FMRP (f4055, Sigma Aldrich), anti-FMRP (ab17722, Abcam) or rabbit monoclonal anti-human IgG antibody (ab109489, Abcam) and left rotating on a wheel for 1 h at room temperature in NT2 Buffer. Beads were then washed in NT2 Buffer and incubated with the diluted lysates in a final volume of 500 μl. Binding was carried out at 4 °C with the samples rotating on a wheel for 2 h. Beads were then washed three times and resuspended in ice-cold NT2. Each sample was split 1/5 for protein and 4/5 for RNA analysis. The protein fraction was resuspended in 1× NuPage LDS (Invitrogen) with 2 mM DTT and left at 70 °C for 20 min. Proteins were run on a 4–12% polyacrylamide gel for 1 h at 160 V. An artificial spike RNA, i.e., an in vitro transcribed RNA fragment derived from the pcDNA3.1 plasmid, was added to the RNA fraction, which was then lysed with 250 μl of TRIzol (Invitrogen) and extracted according to manufacturer's instructions. RNA was analyzed by real-time qRT-PCR with iTaq Universal SYBR Green Supermix (Bio-Rad Laboratories). RIP data analysis was performed as follows. For each target, the mean Ct value from a technical duplicate was normalized to the input RNA fraction (at a 1/10 dilution) Ct value (ΔCt) to account for RNA sample preparation differences, using the equation: $\Delta Ct = [Ct_{RIP\ or\ IgG} - Ct_{input} - Log_2 10]$. The percentage of input was calculated by linear conversion of the normalized ΔCt as $2^{-\Delta Ct}$. This value was then adjusted to take into account the difference in the amplification between the control IgG and the IP fractions using the artificial spike RNA, as follows. We first calculated the spike RNA ΔCt and percentage of input as described above. Then, we normalized the percentage of input of each target using the percentage of input of the spike RNA.

**In vitro binding and competition assays**. In vitro binding assay was performed using in vitro transcribed biotinylated RNA (corresponding to three regions spanning the *HuD* 3′UTR), HeLa extract (containing FMRP protein), and purified recombinant FUS proteins form HeLa, as follows.

For *HuD* 3′UTR fragments, biotinylated RNA preparation was carried out using PCR products generated from hiPSC-FUS^WT gDNA. The forward PCR primers contained the T7 RNA polymerase promoter sequence (T7): TAATACGACTC ACTATAGGG. Primers sequences are listed in Supplementary Table 2. For the RNA negative control, a DNA fragment containing the T7 promoter was obtained by cutting the pSI-Check2 vector with EcoRV and HindIII. These DNA templates were used for in vitro RNA transcription with the T7 polymerase MAXIscript kit (Invitrogen), in presence of 0.2 mM Biotin-16-UTP (Roche). RNA was purified by adding one volume per sample of phenol:chloroform:isoamyl alcohol (25:24:1)

(Thermo Fisher), followed by centrifugation at 4 °C for 10 min at 1000 rpm and precipitation of the upper aqueous phase with ethanol at −80 °C overnight.

For biotin pull-down, Streptavidin MagneSphere paramagnetic particles (Promega) were first washed in EMSA buffer 1× (EMSA buffer 2×: 40 mM Hepes pH 7.9, 150 mM KCl, 3 mM MgCl₂, 2 mM DTT, 10% glycerol with 100 U/ml RNAse inhibitor and 1× protease inhibitor cocktail; Roche) four times. Beads were then incubated in EMSA 1× with 150 μg of E. Coli tRNA at RT for 10 min. After the treatment, the beads were resuspended in EMSA 1× with 100 U/ml RNAse inhibitor and 1× protease inhibitor cocktail (Roche). FRMP binding assay was performed by incubating biotinylated transcripts (250 ng) and 75 μg of HeLa cytoplasmic lysate for 30 min on ice in EMSA buffer 1×.

FUS protein purification for competition assays was performed as follows. Stable and inducible RFP-flag-FUS$^{P525L}$ and RFP-flag-4FL_FUS$^{P525L}$ HeLa lines were induced with 200 ng/μl doxycycline for 24 h. Cells were lysed with RIPA buffer (50 mM Tris-HCl pH 7.5, 150 mM NaCl, 1% NP-40, 0.5% sodium deoxycholate, 0.1% SDS, 1 mM EDTA, 1 mM EGTA and 1× protease inhibitor cocktail; Roche). Anti-FLAG M2 Magnetic Beads (Sigma Aldrich) were washed in Tris Buffer Saline (TBS) two times. A volume containing 5 mg of protein extract was incubated with beads and binding was carried out at 4 °C on a rotating wheel overnight. Beads were then washed in Low buffer (50 mM Tris HCl pH 7.5, 150 mM NaCl, 1 mM EDTA, 5% glycerol, 0,25% NP40) three times and in High buffer (50 mM Tris HCl pH 7.5, 500 mM NaCl, 1 mM EDTA, 5% glycerol, 0,25% NP40) two times. Beads were then resuspended in TBS with Flag peptide (Millipore) in final volume of 50 μl and left at 4 °C on a rotating wheel for 5 minutes. This step was repeated three times, obtaining three different elution samples. Each sample (1/10) was resuspended in 1× NuPage LDS (Invitrogen) with 2 mM DTT and left at 70 °C for 10 min. Proteins were run on a 4–12% polyacrylamide gel for 1 h at 150 V and colored with PageBlue Protein staining solution (Thermo Scientific) overnight.

For competition assay between FMRP and FUS, Streptavidin MagneSphere paramagnetic particles (Promega) were first washed in EMSA buffer 1× four times. Beads were then incubated in EMSA 1× with 150 μg of E. coli tRNA at RT for 10 min and resuspended in EMSA 1× with 100 U/ml RNAse inhibitor and 1× protease inhibitor cocktail (Roche). The assay was then performed in presence of 250 ng of biotinylated transcript, 75 μg of HeLa cytoplasmic lysate and 30 ng of purified FUS protein. RNA-protein complexes were then incubated with streptavidin beads in a final volume of 150 μl. Binding was carried out at 4 °C with on a rotating wheel for 1 h. Beads were then washed in 1× EMSA for three times. Each sample was resuspended in 20 μl PBS with 1× NuPage LDS (Invitrogen) and 2 mM DTT. Complexes were analyzed by western blotting.

**Luciferase assay**. The pSI-Check2 vector containing HuD 3′UTR (RLuc-HuD 3′ UTR)[9] was transfected alone or in combination with epB-Bsd-TT-FMR1 or epB-Bsd-TT-eGFP in 5 × 10⁴ pre-seeded HeLa cells expressing RFP-FUS$^{P525L}$ in a 24-well plate using Lipofectamine 2000 (Life Technologies), following manufacturer's instructions. Cells were harvested 24 h of post-transfection and RLuc and FLuc activities were measured by Dual Glo luciferase assay (Promega), according to the manufacturer's protocol.

**Axon branching and growth upon axotomy assays**. After dissociation at day 5 of differentiation, hiPSC-derived MNs were grown in microfluidics for 7 days. For axon branching analysis, the initial density was 5 × 10⁴ cells per cm². For axotomy experiments, the initial density was 10⁵ cells per cm². Axotomy was performed using three different methods: (1) Trypsin-EDTA 0.25% (Thermo Fisher Scientific) treatment for 15 min (Figs. 4c, 5c, and 8d and Supplementary Fig. 6c); (2) repeated vacuum aspirations (Fig. 4d); (3) Accutase (Thermo Fisher Scientific) treatment for 15 min (Fig. 4e). The axon chamber was reperfused with PBS until effective removal of the damaged axons, without disturbing the cell bodies in the soma compartment.

Quantitative analyses of axon branches and branch points were conducted with Skeleton, a plugin of Fiji[74]. Thirty hours after axotomy, immunofluorescence staining with an anti-TUJ1 antibody was performed as described in the "Immunofluorescence" section.

**Statistics and reproducibility**. Statistical analysis, graphs, and plots were generated using GraphPad Prism 6 (GraphPad Software). As indicated in each figure legend, Student's t-test or ordinary one-way ANOVA was performed, and data set are shown in dot plots indicating mean ± standard deviation (st.dev.) or standard error of the mean (s.e.m.). Sample size and the definition of replicates for each experiment is also indicated in the figure legends. p values are indicated in the figures.

**Reporting summary**. Further information on research design is available in the Nature Research Reporting Summary linked to this article.

## Data availability

All data generated or analyzed during this study are included in this published article and its supplementary information files. Newly generated plasmids are available upon request. Source data underlying graphs and charts presented in the figures are provided in Supplementary Data 2.

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

## Acknowledgements

The authors wish to thank the Imaging Facility at Center for Life Nano Science, Istituto Italiano di Tecnologia, for support and technical advice. We are grateful to Jared Sterneckert (Technische Universität Dresden, Germany) for sharing their hiPSC lines. We thank Riccardo De Santis (The Rockefeller University, NY, USA), Serena Carra (University of Modena and Reggio Emilia, Italy) and Silvia Di Angelantonio and Irene Bozzoni (Sapienza University of Rome, Italy) for helpful discussion. This work was partially supported by: Sapienza University, Fondazione Istituto Italiano di Tecnologia and a grant from Istituto Pasteur Italia—Fondazione Cenci Bolognetti (to A.R.); UK Medical Research Council (MR/M008606/1 and MR/S006508/1 to P.F.; MR/S022708/1 to E.M.C.F. and P.F.); Motor Neurone Disease Association (885-792 to P.F. and T.C.); NIHR-UCLH Biomedical Research Centre to P.F.); Rosetrees Trust (to E.M.C.F.).

## Author contributions

Conceptualization, M.G.G., M.R. and A.R.; Formal analysis, M.G.G., N.B., M.R.; Investigation, M.G.G., N.B. and F.S.; Methodology, M.G.G., F.S., Michela Mochi, V.d.T., R.R.N., M.R, Mariangela Morlando; Project administration, A.R.; Supervision, A.R., P.F., E.M.C.F. and T.J.C.; Writing—original draft, A.R. All authors read and approved the final manuscript.

## Competing interests

The authors declare no competing interests.
