## [Peer Review File · Communications Biology]

Reviewers' comments:

Reviewer #1 (Remarks to the Author):

This paper is the extension of a previous study reporting FUS P525L binds to 3'UTR of HuD and stabilizes HuD mRNA leading to increased HuD protein levels (De Santis et al. Cell Reports 2019). First, the authors verify their previous findings that HuD mRNA and protein levels are upregulated in FUS P525L expressing motor neurons compared to FUS WT motor neurons in both human iPSCs and mice. The authors further confirm this finding by examining the downstream targets of HuD including NRN1 and GAP43. The authors newly report that FMRP also binds to HuD 3'UTR and repress its translation, but in the presence of FUS P525L, the mutant FUS outcompetes FMRP and thereby leading to increased HuD levels. They further show that FUS P525L motor neurons show aberrant axonal phenotypes compared to those expressing FUS WT. The authors further investigate how increased HuD levels lead to aberrant axonal phenotypes in FUS P525L motor neurons. They found that aberrant axonal phenotypes could be due to increased NRN1 levels because knockdown of NRN1 ameliorated the phenotypes in FUS P525L motor neurons.

The authors present interesting findings on potential mechanism by which FUS P525L lead to motor neuron phenotypes. This manuscript would be suitable for publication if the following comments are addressed.

Major points:

1. Clear evidence would be necessary to conclude that FUS P525L competes with FMRP on HuD 3'UTR. Do both FUS P525L and FMRP bind to the same binding sequence in HuD 3'UTR? In vitro competition binding assay with purified FUS P525L and FMRP would be helpful.
2. Figure 2E was conducted only in the presence of FUS P525L with or without FMRP. Other controls are missing. Particularly, it would be important to know the luciferase activity upon overexpression of FMRP alone, or that compared to FUS WT.
3. In Figures 1C, 5A and 6D, the authors used motor neurons that express FUS WT alone, FUS P525L alone or both FUS WT and HuD. The authors should clearly state how the motor neurons were established because it is questionable whether motor neurons that express FUS WT alone or FUS P525L alone could be served as proper controls for those that express FUS WT together with HuD. For the FUS WT+ HuD motor neurons, it is stated that SYN1::HuD construct is added to the FUS WT motor neurons. If this is the case, then for example SYN1::GFP construct should be added to FUS WT or FUS P525L motor neurons to compare with FUS WT + HuD. Otherwise, it would be difficult to interpret the findings.

Minor points:

1. Was the experiment Figure 2B for FUS WT and FUS P525L conducted simultaneously? If so, to compare the result of FUS WT and P525L side by side, organizing the data in one graph would help understand better.
2. Figure 2C, GAP43 blot, what is the upper band that is found in both FMRP WT and KO?
3. Figure 7A, please reveal all the data points and show statistical significance.
4. The manuscript requires extensive edits to clearly convey the results. Throughout the manuscript, please correct spelling errors (e.g. page 5, line 15, "synthetised"), redundant words (e.g. page 5, line 14, "using and") and sentences without evidence (e.g. page 2, line 6).
5. When referring to Fus-delta14 knock-in mouse model (reference 33), the statement "in which a

frameshift mutation leads to a complete loss of the nuclear localization signal (NLS)" should be carefully rewritten because the authors of this paper reported that they do not see nuclear depletion of mutant FUS.

6. Figure 2B, it is not clear what the y-axis referring to. If this is the RIP result of Figure 2A, should it be "relative enrichment of the mRNAs pulled down by FMRP"?

7. In the abstract, page 2 line 6 "cytoplasmic localization of mutant FUS leads to upregulation of HuD levels through competition with FMRP for HuD 3'UTR binding." This sentence should be restated because previous papers (references 9 and 33) showed that mutant FUS (P525L or NLS deletion mutant) does not completely localize to the cytoplasm. To provide evidence that FUS P525L binds to HuD 3'UTR in the cytoplasm, immunostaining of mutant FUS combined with FISH for HuD mRNA would be required.

8. The title could be modified to reflect the main findings of the paper and be more specific and active instead of general/passive.

9. Abstract should be rewritten as this paper focuses on certain RBPs, not a network of RBPs.

Reviewer #2 (Remarks to the Author):

The authors present the interesting finding that ALS associated mutations in FUS increase HuD levels by decreasing FMRP levels, which in turn results in increased translation of Nrn1 and (less so) Gap43 mRNAs to increase axon growth. The authors bring together several different approaches as well as human iPSC and mouse samples to conclude this. Together, these findings would bring new insight into mechanism(s) underlying previous reports for increased axon growth from FUS mutant neurons. Overall, the studies are well done and results clear. However, I think the conclusions are not fully supported or flushed out by the authors and they miss citing a few critical publications, including some that may provide further insight into their results.

1. Based on effects of FMRP KO on HuD levels, the authors conclude that translation of HuD mRNA is increased with mutant FUS mutations. There clearly is an increase in HuD protein with FMRP KO, but I do not see how the authors distinguish effects of elevated HuD mRNA in the FUS mutants (i.e., miRNA effects) vs. translational regulation of the mRNA. It is confusing that there is no synergistic effect miRNA effect plus loss of FMRP translational suppression of HuD mRNA. At least, this is not clear from the data as presented. Another complicating factor that is not discussed but shown in Suppl Table 2 is that HuD (ELAVL4) binds to its own mRNA, and the authors do not discuss this possibility (which presumably would stabilize the mRNA, further complicating how to distinguish translational upregulation vs. more translation because of more HuD mRNA present). Also, the authors suggest that FMRP directly binds to 3'UTR of HuD mRNA (pg 6), but they have not shown this.

2. The authors indicate that axons grow longer in FUS mutant neurons, both initial extension and regeneration in vitro, but no quantifications are provided for this. The images certainly support this

conclusion, but these are single images. The authors' conclusions would be strengthened by providing quantitation across multiple replicates.

3. On pg 7, the authors point to effects of HuD overexpression on neurite outgrowth in PC12 cells and cortical neurons – they should consider referencing Perrone-Bizzozero et al. ASN Neuro 3, 259 that showed increased axon growth in vivo in HuD overexpressor mice.

4. The effects on Gap43 mRNA and protein are overall modest compared to the effect on Nrn1. Gomes et al. J Cell Sci 130, 3650 showed that HuD binds stabilizes Nrn1 mRNA in vitro and that Nrn1's 3'UTR ARE has a higher binding affinity for HuD than does Gap43's ARE. Further, Nrn1 ARE was able to displace HuD from Gap43. This may explain the modest effects on Gap43 mRNA.

5. Notably, the Akten paper cited by the authors did not show that HuD binds to Nrn1 mRNA's ARE, but rather that Nrn1 mRNA & HuD colocalize and co-precipitate. The authors may also want to mention the axonal localization of Nrn1 mRNA in the discussion – note that Merianda et al. J Neurosci 33, 13735 reported that the axonally localizing Nrn1 mRNA with both its 5' and 3' UTR optimally promoted axon growth in DRG neuron cultures. Though the DRGs seem to use the 5'UTR for axonal localization while CNS neurons used the 3'UTR that contains the ARE.

6. Finally, I think that the authors need to be very clear to the reader when they are referring to mRNA vs. protein. They are proposing a complicated mechanism and I got lost within the text at times. For example, within the abstract 'Mechanistically, cytoplasmic localization of mutant FUS leads to upregulation of HuD protein levels through competition with FMRP for HuD mRNA 3'UTR binding' would provide more clarity for the reader not accustomed to thinking of RNA protein interactions. Similarly, on page 5 line 11, the authors should indicate that 'HuD mRNA puncta' as RNA binding proteins often also appear punctate. Consistent text formatting with mRNA in italics and protein in standard font would help.

Minor points:

1. The authors jump between figures, with data introduced in figures but not discussed included in the results for several sections (e.g., Nrn1 & Gap43 in Fig 2 that is not mentioned at all on pg 6)

2. pg 3, line 10 'pathological mechanism' – mechanism of what? (I presume for ALS)

3. pg 4, line 4, 'aberrant branching & axonal outgrowth' – it would be clearer if the authors indicated whether it is increased or decreased.

4. pg 4, line 23 – 'mutant FUS expression to lead'.

5. Figures in general – the graphs provided are difficult to distinguish data points from average/SEM lines. All of the figures with those graph panels also include panels with color images, so the authors' data would be easier to interpret with colors to distinguish the data points, averages, and errors. Another minor point is lack of error calculations for the control samples – even if the authors are setting the controls at 1, there certainly must be some error associated with the measurements and this should be reported.

6. Figure 6 – Given the number of publications pointing to GAP43's role at the growth cone, the

authors should provide high magnification images of axon terminals. Also, with its concentration there and the GAP43 mRNA localizing to distal axons, the authors might consider showing quantification of GAP43 levels from growth cone/terminal axon images.

7. Figure 8 – The authors should be more specific than ‘aberrant’, which could be increased or decreased. Also, there is the open question on how HuD binding may affect its own mRNA.

8. Suppl Figure S7 – The authors should be clear whether or not the full 3’ and 5’UTRs of HuD are included or not in these expression constructs rather than just the poly A signal.

Reviewer #3 (Remarks to the Author):

Brief Summary

In this manuscript, the authors are building on previous work showing that FUS mutations can change the translation of key factors in axon outgrowth and axon regeneration, and they provide a plausible mechanism for how this could occur through the negative regulation of FMRP binding to the HuD 3’ UTR.

Overall Impression

The authors do a good job in supporting their proposed model with experiments looking at transcription, translation, and phenotypes in axon growth assays. Generally, their quantifications are reasonable, though some need additional biological replicates as mentioned below. Images aid the reader in seeing the phenotypes described are robust. Some additional data and quantification are required to fully support the conclusions and to put the authors’ findings into the context of ALS.

1. Could you please clarify the nature of the P525L mutation early on in the manuscript? Does this disrupt the NLS or other important domains? A diagram showing the two mutations being used in the paper might be of assistance to readers. Also, how much protein remains in delta14 mutant mice? The authors say there is a deletion of just the NLS.

2. It’s very unclear to me what Figure S1 is trying to show. Figures are blurry, and while the authors make mention of the conservation of the 3’ UTRs, it’s unclear what I should be looking at on each screenshot. More clarification and better images would be helpful here.

3. In Figure 3A, it would be nice to have images matching those in Figure 3C of the axons that were initially grown from these two genotypes. It is confusing what is meant by “junctions.” Is this branch points, or crossing between two axon branches? Please define this better, or perhaps just call them “branch points.” The axons in 3C are a result of regeneration, rather than initial outgrowth, and I believe this type of regrowth can look much different than initial axon outgrowth. Thus, a visual comparison would be helpful. Finally, how are the authors able to investigate these parameters if individual neuron branches cannot be clearly separated? From the images, this looks impossible. Please add explanation or your criteria for selecting certain axons to be counted.

4. More images and quantification is needed in Figure 7 to sufficiently be convinced of the result. For 7a, please do biological replicates (three separate wells of each condition, for example) and perform the statistical analysis on this; technical replicates on a single sample of each conditions are not sufficient. For 7B, please show traces of representative axons in the experiment. For 7C, why do you not see the robust regeneration phenotypes shown in previous figures in the top condition? Images are not convincing in this case. I would suggest that authors quantify three biological replicates and show quantification as in prior figures.

5. The model fits with the authors’ data and I believe that the authors have provided good support

for most of the model. The one piece where I'm a bit unsure is the part where FMRP and FUS compete for the same site on the 3' UTR. The luciferase assay provided by the authors is a solid start and, when combined with the FMRP RIP assay, suggests their model is likely. A more convincing demonstration of this part of the mechanism would help to strengthen the paper. The authors say that the HuD 3'UTR contains multiple possible regulatory sites. For example, could you delete the predicted binding site and eliminate both FMRP RIP and FUS binding? Alternatively, could you test whether FUS variants can be IP'ed for the RIP assay and evaluated for HuD binding in the presence or absence of FMRP?

6. I am left after reading the paper with a bit of disorientation about how this is all connected to ALS. It needs to be made more clear, with citations, why exuberant regeneration could contribute to ALS. It's also not entirely clear what is the evidence for extra axon branching in patients with this mutation in the first place? In doing some reading, the 525L mutation is associated with severe, juvenile ALS variants. The authors note that axon branching increases and outgrowth increases are common across many ALS causing mutations. Can the authors compare (literature is fine) the effect on primary outgrowth, or regenerative outgrowth, among different reports studying different ALS mutations? It would help to put these findings into an appropriate context.

Garone et al.
Point-by-point response to the referees' comments

We are grateful to the Editor and to the 3 reviewers for their constructive criticisms and suggestions, which have contributed to improve the paper. Please note that we have added an author, Prof. Mariangela Morlando, who contributed to the revision. One main figure and 3 supplementary figures have been added to accommodate new data.

EDITOR

In particular, please note that the following revisions would be necessary for us to contact our referees again:

- perform FUS P525L and FMRP competition binding assay;

We have performed this experiment including the results in the new Figure 2 (note that the old Figure 2 has been split in new Figures 2 and 3 to accommodate this new data).

- include specific control samples for experiment presented in Fig. 2E;

Requested controls have been added. See Reviewer #1, point 2, for details.

- provide detailed description of how motor neurons were established;

A detailed description of the protocols used for establishing motor neurons is provided in the MATERIALS AND METHODS section. We have better specified in the RESULTS the method used for hiPSC differentiation, as follows (page 5, lines 5-8):

"In order to gain insights into HuD regulation in ALS, we took advantage of spinal MNs derived from isogenic pairs of FUS WT and P525L hiPSC lines (hereafter FUS^{WT} and FUS^{P525L}) by inducible expression of a "programming module" consisting of the transcription factors Ngn2, Isl1 and Lhx3 (NIL) [31,32]."

We have also provided a more detailed description of how the SYN::HuD MNs have been established as requested by Reviewer #1 (see response to point 3).

- include additional quantification of the axon growth assay proposed by reviewer 2;

Requested quantifications have been added to all the axon assays. See Reviewer #2, point 2, for details.

- include biological replicates for the experiments presented in Fig. 7.

Biological replicates have been added to Fig. 7 as requested. See Reviewer #3, point 4, for details.

Reviewer #1 (Remarks to the Author):

This paper is the extension of a previous study reporting FUS P525L binds to 3'UTR of HuD and stabilizes HuD mRNA leading to increased HuD protein levels (De Santis et al. Cell Reports 2019). First, the authors verify their previous findings that HuD mRNA and protein levels are upregulated in FUS P525L expressing motor neurons compared to FUS WT motor neurons in both human iPSCs

and mice. The authors further confirm this finding by examining the downstream targets of HuD including NRN1 and GAP43. The authors newly report that FMRP also binds to HuD 3'UTR and repress its translation, but in the presence of FUS P525L, the mutant FUS outcompetes FMRP and thereby leading to increased HuD levels. They further show that FUS P525L motor neurons show aberrant axonal phenotypes compared to those expressing FUS WT. The authors further investigate how increased HuD levels lead to aberrant axonal phenotypes in FUS P525L motor neurons. They found that aberrant axonal phenotypes could be due to increased NRN1 levels because knockdown of NRN1 ameliorated the phenotypes in FUS P525L motor neurons.

The authors present interesting findings on potential mechanism by which FUS P525L lead to motor neuron phenotypes. This manuscript would be suitable for publication if the following comments are addressed.

Major points:

1. Clear evidence would be necessary to conclude that FUS P525L competes with FMRP on HuD 3'UTR. Do both FUS P525L and FMRP bind to the same binding sequence in HuD 3'UTR? In vitro competition binding assay with purified FUS P525L and FMRP would be helpful.

We now provide this important evidence, as also requested by the other reviewers (Ref. 2 point 1 and Ref. 3 point 5). Specifically, we have performed an in vitro competition assay with in vitro transcribed RNA corresponding to 3 regions of the HuD 3'UTR. We found that the purified FUS-P525L protein competes with FMRP on one of these regions (comparing FUS-P525L with a mutant derivative in which RNA binding is impaired; see Daigle et al., 2013). These results have been included in the new Figure 2 (note that the old Figure 2 has been split in new Figures 2 and 3 to accommodate this new data).

2. Figure 2E was conducted only in the presence of FUS P525L with or without FMRP. Other controls are missing. Particularly, it would be important to know the luciferase activity upon overexpression of FMRP alone, or that compared to FUS WT.

As requested by the referee we have repeated this experiment by overexpressing FMRP in presence of FUS-WT or RFP alone (since both FUS-P525L and FUS-WT transgenes were tagged with RFP). As shown in the new Figure 3C, FMRP overexpression strongly reduced the luciferase activity in presence of both RFP and RFP-FUS-WT, while such reduction was lower in presence of RFP-FUS-P525L, in agreement with our model.

3. In Figures 1C, 5A and 6D, the authors used motor neurons that express FUS WT alone, FUS P525L alone or both FUS WT and HuD. The authors should clearly state how the motor neurons were established because it is questionable whether motor neurons that express FUS WT alone or FUS P525L alone could be served as proper controls for those that express FUS WT together with HuD. For the FUS WT+ HuD motor neurons, it is stated that SYN1::HuD construct is added to the FUS WT motor neurons. If this is the case, then for example SYN1::GFP construct should be added to FUS WT or FUS P525L motor neurons to compare with FUS WT + HuD. Otherwise, it would be difficult to interpret the findings.

We have better specified in the RESULTS section how MNs used in these experiments have been established (page 10, lines 1-5). Moreover, as suggested by the referee, we have repeated the analysis of HuD, NRN1 and GAP43 expression levels in FUS^{WT} motor neurons carrying a SYN1::RFP construct, which served as a proper control of SYN1::HuD containing cells. As shown in the new Figure 5A, expression levels of the three genes of interest are not significantly

changed in FUS^{WT} + SYN1::RFP control motor neurons compared to FUS^{WT}, thus excluding aspecific effects of the SYN1 promoter-based construct.

Minor points:

1. Was the experiment Figure 2B for FUS WT and FUS P525L conducted simultaneously? If so, to compare the result of FUS WT and P525L side by side, organizing the data in one graph would help understand better.

These experiments had been indeed conducted simultaneously. Following the referee's suggestion, we have now organized the data in a single graph.

2. Figure 2C, GAP43 blot, what is the upper band that is found in both FMRP WT and KO?

Two bands (MW: 38 kDa and 43 kDa) have been previously reported in GAP43 WB of brain tissue (see for instance, Kim et al., 2019; <https://doi.org/10.5607/en.2019.28.1.85>). see also the snapshot of the datasheet of a commercial antibody shown below (<https://www.cellsignal.com/products/primary-antibodies/gap43-d9c8-rabbit-mab/8945>). The quantification shown in the figure has been accordingly performed on both bands.

3. Figure 7A, please reveal all the data points and show statistical significance.

As also requested by reviewer #3, comment 4, this experiment has been now repeated to provide statistical significance (n=3 biological replicates).

4. The manuscript requires extensive edits to clearly convey the results. Throughout the manuscript, please correct spelling errors (e.g. page 5, line 15, “synthetised”), redundant words (e.g. page 5, line 14, “using and”) and sentences without evidence (e.g. page 2, line 6).

The manuscript has been edited as suggested.

5. When referring to Fus-delta14 knock-in mouse model (reference 33), the statement “in which a frameshift mutation leads to a complete loss of the nuclear localization signal (NLS)” should be carefully rewritten because the authors of this paper reported that they do not see nuclear depletion of mutant FUS.

Also in response to Reviewer #3, major point 1, this sentence has been rephrased as follows (page 5, lines 10-13):

“In parallel, we used the Fus-Δ14 knock-in mouse model, in which a frameshift mutation leads to the loss of the C-terminal nuclear localization signal (NLS) (Supplementary Figure S2A), causing partial mislocalization to the cytoplasm without altering total Fus protein levels (Supplementary Figure 2B) [33].”

6. Figure 2B, it is not clear what the y-axis referring to. If this is the RIP result of Figure 2A, should it be “relative enrichment of the mRNAs pulled down by FMRP”?

The figure legend has been changed following reviewer’s suggestion.

7. In the abstract, page 2 line 6 “cytoplasmic localization of mutant FUS leads to upregulation of HuD levels through competition with FMRP for HuD 3’UTR binding.” This sentence should be restated because previous papers (references 9 and 33) showed that mutant FUS (P525L or NLS deletion mutant) does not completely localize to the cytoplasm. To provide evidence that FUS P525L binds to HuD 3’UTR in the cytoplasm, immunostaining of mutant FUS combined with FISH for HuD mRNA would be required.

We have eliminated the mention to cytoplasmic localization and restated the sentence as follows (page 2, lines 6-7):

“mutant FUS leads to upregulation of HuD protein levels through competition with FMRP for *HuD* mRNA 3’UTR binding.”

8. The title could be modified to reflect the main findings of the paper and be more specific and active instead of general/passive.

9. Abstract should be rewritten as this paper focuses on certain RBPs, not a network of RBPs.

We have modified title and abstract to better and more specifically reflect the main findings, as suggested:

TITLE: “ALS-FUS mutation affects the activities of HuD/ELAVL4 and FMRP leading to axon phenotypes in motoneurons”

ABSTRACT: “Mutations in the RNA-binding protein (RBPs) FUS have been genetically associated with the motoneuron disease Amyotrophic Lateral Sclerosis (ALS). Using both human induced Pluripotent Stem Cells and mouse models, we found that FUS-ALS causative mutations affect the activity of other two relevant RBPs with important roles in neuronal RNA metabolism: HuD/ELAVL4 and FMRP. Mechanistically, mutant FUS leads to upregulation of HuD protein levels through competition with FMRP for *HuD* mRNA 3’UTR binding. In turn, increased HuD levels overly stabilize the transcript levels of its targets, NRN1 and GAP43. As a consequence, mutant FUS motoneurons show increased axon branching and growth upon injury, which could be rescued by dampening NRN1 levels. Since similar phenotypes have been previously described in SOD1 and TDP-43 mutant models, aberrant axonal growth and branching might represent broad early events in the pathogenesis of ALS.”

Reviewer #2 (Remarks to the Author):

The authors present the interesting finding that ALS associated mutations in FUS increase HuD levels by decreasing FMRP levels, which in turn results in increased translation of Nrn1 and (less so) Gap43 mRNAs to increase axon growth. The authors bring together several different approaches as well as human iPSC and mouse samples to conclude this. Together, these findings would bring new insight into mechanism(s) underlying previous reports for increased axon growth from FUS mutant neurons. Overall, the studies are well done and results clear. However, I think the conclusions are not fully supported or flushed out by the authors and they miss citing a few critical publications, including some that may provide further insight into their results.

1. Based on effects of FMRP KO on HuD levels, the authors conclude that translation of HuD mRNA is increased with mutant FUS mutations. There clearly is an increase in HuD protein with FMRP KO, but I do not see how the authors distinguish effects of elevated HuD mRNA in the FUS mutants (i.e., miRNA effects) vs. translational regulation of the mRNA. It is confusing that there is no synergistic effect miRNA effect plus loss of FMRP translational suppression of HuD mRNA. At least, this is not clear from the data as presented.

Collectively, in our previous (refs. [9] and [17]) and present work we found that:

- In MNs, both FMRP and miR-375 are expressed and both mRNA and protein levels of HuD/ELAVL4 are increased

- In MNs, HuD/ELAVL4 mRNA levels change in response to miR-375 (figure 5D in ref. [17]) but do not change in response to FMRP (figure 3B in present work); while HuD/ELAVL4 protein levels change in response to FMRP (figure 3A in present work)

- In HeLa, where FMRP is expressed but miR-375 is not (figure Suppl. 3B-C in ref. [9]), the HuD/ELAVL4 3'UTR luciferase reporter mRNA is not changed in presence of FUS^{P525L}, while the luciferase activity is increased (figure 2C-D in ref. [9])

We can conclude that in mutant FUS MNs, increased HuD/ELAVL4 mRNA levels are due to decreased miR-375 levels, while increased HuD/ELAVL4 protein levels can be due to the double effect of miR-375 decrease and FMRP displacement by mutant FUS.

We have better clarified this point in the DISCUSSION as follows (page 11, lines 20-23):

“Since *HuD* mRNA levels are affected by miR-375 [9] but not by FMRP (present work), we can conclude that in mutant FUS MNs increased *HuD/ELAVL4* mRNA levels are due to decreased miR-375 expression [17], while increased HuD/ELAVL4 protein levels can be due to the double effect of the loss of both miR-375 and of FMRP regulation.

Another complicating factor that is not discussed but shown in Suppl Table 2 is that HuD (ELAVL4) binds to its own mRNA, and the authors do not discuss this possibility (which presumably would stabilize the mRNA, further complicating how to distinguish translational upregulation vs. more translation because of more HuD mRNA present).

HuD autoregulation mediated by 3'UTR binding has been previously shown in animal models (refs: Samson, 1998; Bolognani et al., 2009; see also Borgeson and Samson, 2005 doi: 10.1093/nar/gki942). As correctly pointed out by the reviewer, an interesting result of our *catRAPID* prediction is that this could be the case also in human. As for the outcome of this binding, *Drosophila* ELAV has been shown to downregulate, rather than stabilize, the expression of its own mRNA (Samson, 1998). Analysis of a possible analogous autoregulatory mechanism in human was beyond the scope of our paper, but we have added a paragraph in the DISCUSSION to mention this possibility (page 11, lines 25-26; and page 12, lines 1-2):

“In addition to miR-375 and FMRP, our *catRAPID* analysis indicates that the HuD 3'UTR might be also a target of the HuD protein (Supplementary Table S2), thus suggesting possible conservation in human of the autoregulatory mechanism previously proposed in *Drosophila* and mouse [47,48].”

Also, the authors suggest that FMRP directly binds to 3'UTR of HuD mRNA (pg 6), but they have not shown this.

We now provide this important evidence, as also requested by the other reviewers (Ref. 1 point 1 and Ref. 3 point 5). Specifically, we have performed an in vitro competition assay with in vitro transcribed RNA corresponding to 3 regions of the HuD 3'UTR. We found that the purified FUS-P525L protein competes with FMRP on one of these regions (comparing FUS-P525L with a

mutant derivative in which RNA binding is impaired; see Daigle et al., 2013). These results have been included in the new Figure 2 (note that the old Figure 2 has been split in new Figures 2 and 3 to accommodate this new data).

2. The authors indicate that axons grow longer in FUS mutant neurons, both initial extension and regeneration in vitro, but no quantifications are provided for this. The images certainly support this conclusion, but these are single images. The authors' conclusions would be strengthened by providing quantitation across multiple replicates.

Axon length has been quantified in multiple biological replicates (see revised figures 4, 5 and 8). This analysis showed significant differences between FUS mutant and WT MNs and in MNs treated with control and NRN1 siRNAs, confirming the conclusions.

3. On pg 7, the authors point to effects of HuD overexpression on neurite outgrowth in PC12 cells and cortical neurons – they should consider referencing Perrone-Bizzozero et al. ASN Neuro 3, 259 that showed increased axon growth in vivo in HuD overexpressor mice.

We have added this relevant reference.

4. The effects on Gap43 mRNA and protein are overall modest compared to the effect on Nrn1. Gomes et al. J Cell Sci 130, 3650 showed that HuD binds stabilizes Nrn1 mRNA in vitro and that Nrn1's 3'UTR ARE has a higher binding affinity for HuD than does Gap43's ARE. Further, Nrn1 ARE was able to displace HuD from Gap43. This may explain the modest effects on Gap43 mRNA.

We have mentioned this important point in the revised DISCUSSION as follows (page 13, lines 4-8):

"The neurotrophic factor NRN1 is one of the primary HuD targets in MNs. HuD stabilizes NRN1 mRNA via AU-rich element (ARE) binding on its 3'UTR [41,59]. Overall, we observed stronger effects on NRN1 compared to GAP43, at both mRNA and protein levels (with the relevant exception of GAP43 at the growth cone). This is in agreement with previous findings showing that the NRN1 ARE has a higher binding affinity for HuD compared to GAP43 ARE [60]."

5. Notably, the Akten paper cited by the authors did not show that HuD binds to Nrn1 mRNA's ARE, but rather that Nrn1 mRNA & HuD colocalize and co-precipitate. The authors may also want to mention the axonal localization of Nrn1 mRNA in the discussion – note that Merianda et al. J Neurosci 33, 13735 reported that the axonally localizing Nrn1 mRNA with both its 5' and 3' UTR optimally promoted axon growth in DRG neuron cultures. Though the DRGs seem to use the 5'UTR for axonal localization while CNS neurons used the 3'UTR that contains the ARE.

A reference to this work has been added to the revised DISCUSSION as follows (page 13, lines 22-24):

"Moreover, axonal localization of Nrn1 mRNA, which is induced after nerve injury and is mediated by the 3'UTR in central nervous system and by the 5'UTR in peripheral nervous system axons, promotes axon growth [68]."

6. Finally, I think that the authors need to be very clear to the reader when they are referring to mRNA vs. protein. They are proposing a complicated mechanism and I got lost within the text at times. For example, within the abstract 'Mechanistically, cytoplasmic localization of mutant FUS leads to upregulation of HuD protein levels through competition with FMRP for HuD mRNA 3'UTR binding' would provide more clarity for the reader not accustomed to thinking of RNA protein interactions. Similarly, on page 5 line 11, the authors should indicate that 'HuD mRNA puncta' as

RNA binding proteins often also appear punctate. Consistent text formatting with mRNA in italics and protein in standard font would help.

We have modified the text as suggested.

Minor points:

1. The authors jump between figures, with data introduced in figures but not discussed included in the results for several sections (e.g., Nrn1 & Gap43 in Fig 2 that is not mentioned at all on pg 6)

We have now included a discussion on the change in NRN1 and GAP43 levels upon FMRP KO in the text referred to these results, as follows (page 7; lines 10-12):

“In FMRP^{KO} MNs we also observed higher transcript and protein levels of two HuD target genes, *NRN1* and *GAP43* (Figure 3A-B).”

2. pg 3, line 10 ‘pathological mechanism’ – mechanism of what? (I presume for ALS)

Correct. This has been specified

3. pg 4, line 4, ‘aberrant branching & axonal outgrowth’ – it would be clearer if the authors indicated whether it is increased or decreased.

We have indicated that is “increased”.

4. pg 4, line 23 – ‘mutant FUS expression to lead’.

Done

5. Figures in general – the graphs provided are difficult to distinguish data points from average/SEM lines. All of the figures with those graph panels also include panels with color images, so the authors’ data would be easier to interpret with colors to distinguish the data points, averages, and errors.

Another minor point is lack of error calculations for the control samples – even if the authors are setting the controls at 1, there certainly must be some error associated with the measurements and this should be reported.

We have modified the graphs including colored data points. Regarding the second minor point, we set the controls at 1 every time individual replicate experiments have been analyzed in separate gels or real-time PCR runs. In these cases, we normalize each experimental condition over the corresponding control (e.g. in fig. 1A, FUS^{P525L} MNs over FUS^{WT} MNs from the same differentiation experiment).

6. Figure 6 – Given the number of publications pointing to GAP43’s role at the growth cone, the authors should provide high magnification images of axon terminals. Also, with its concentration there and the GAP43 mRNA localizing to distal axons, the authors might consider showing quantification of GAP43 levels from growth cone/terminal axon images.

We thank the reviewer for this suggestion. We now provide IF analysis of the growth cone, showing a remarkable difference in the amount of GAP43, which is absent in FUS-WT cells and clearly detected in FUS mutant cells. These results are shown in the new Figure 7D (we have moved the old panels in Supplementary fig. S7).

7. Figure 8 – The authors should be more specific than ‘aberrant’, which could be increased or decreased. Also, there is the open question on how HuD binding may affect its own mRNA.

We have specified throughout the text that we mean “increased”. For what concerns HuD autoregulation, see response to point 1.

8. Suppl Figure S7 – The authors should be clear whether or not the full 3’ and 5’UTRs of HuD are included or not in these expression constructs rather than just the poly A signal. **These constructs are devoid of both UTRs. This has been specified in the MATERIALS AND METHODS, “Plasmids construction” section, as follows (page 15, lines 13-14): “Such HuD sequence is devoid of both 5’ and 3’UTR”.**

Reviewer #3 (Remarks to the Author):

Brief Summary

In this manuscript, the authors are building on previous work showing that FUS mutations can change the translation of key factors in axon outgrowth and axon regeneration, and they provide a plausible mechanism for how this could occur through the negative regulation of FMRP binding to the HuD 3’ UTR.

Overall Impression

The authors do a good job in supporting their proposed model with experiments looking at transcription, translation, and phenotypes in axon growth assays. Generally, their quantifications are reasonable, though some need additional biological replicates as mentioned below. Images aid the reader in seeing the phenotypes described are robust. Some additional data and quantification are required to fully support the conclusions and to put the authors’ findings into the context of ALS.

1. Could you please clarify the nature of the P525L mutation early on in the manuscript? Does this disrupt the NLS or other important domains? A diagram showing the two mutations being used in the paper might be of assistance to readers. Also, how much protein remains in delta14 mutant mice? The authors say there is a deletion of just the NLS.

A diagram showing the mutations used in the paper has been added to Supplementary Fig. S2. The P525L mutation has been now introduced in the INTRODUCTION and at the beginning of the RESULTS section as follows:

INTRODUCTION (page 3, lines 6-9): “In the RBP FUS the most severe ALS mutations, including the P525L, lie within its C-terminal nuclear localization signal (PY-NLS domain), impairing the interaction with the nuclear import receptor Transportin-1 (TNPO1) and reducing nuclear localization [2].”

RESULTS (page 5, lines 8-10): “The P525L mutation, localized in the PY-NLS domain (Supplementary Figure S2A), causes severe mislocalization of the FUS protein in the cytoplasm and is often associated to juvenile ALS [2].”

For what concerns Fus protein levels in the mouse model, we have previously shown that the wild-type and Delta14 Fus alleles produce equal amounts of Fus protein (ref. [33]). New quantification of WT and mutant Fus in mouse brain has been added in Supplementary Fig. S2. Also in response to Reviewer#1, minor point 5, we have better specified the consequences of the delta14 mutation as follows (page 5, lines 10-13):

“In parallel, we used the Fus-Δ14 knock-in mouse model, in which a frameshift mutation leads to the loss of the C-terminal nuclear localization signal (NLS) (Supplementary Figure S2A), causing partial mislocalization to the cytoplasm without altering total Fus protein levels (Supplementary Figure 2B) [33].”

2. It's very unclear to me what Figure S1 is trying to show. Figures are blurry, and while the authors make mention of the conservation of the 3' UTRs, it's unclear what I should be looking at on each screenshot. More clarification and better images would be helpful here.

For clarity, we have decided to keep in Figure S1 only the graph showing the comparison of the evolutionary conservation (phyloP100way score) between the 3'UTRs of the indicated genes, and a better image of the 3'UTR of ELAVL4/HuD, in which we have indicated the position of the miR-375 binding sites.

3. In Figure 3A, it would be nice to have images matching those in Figure 3C of the axons that were initially grown from these two genotypes.

We have included representative pictures of immunostained (TUBB3) axons and the respective ImageJ Skeleton analysis in Supplementary Fig. S6A.

It is confusing what is meant by "junctions." Is this branch points, or crossing between two axon branches? Please define this better, or perhaps just call them "branch points."

We thank the referee for the suggestion, we have now indicated the crossing points as "branch points".

The axons in 3C are a result of regeneration, rather than initial outgrowth, and I believe this type of regrowth can look much different than initial axon outgrowth. Thus, a visual comparison would be helpful.

A direct comparison of regenerated and untreated axons is shown in Supplementary Fig. S6C as immunofluorescence images. We have also included brightfield images of the axons just before axotomy in Supplementary Fig. S6B.

Finally, how are the authors able to investigate these parameters if individual neuron branches cannot be clearly separated? From the images, this looks impossible. Please add explanation or your criteria for selecting certain axons to be counted.

Branches and branch points were investigated in cells plated at low density for the specific purpose of analyzing single cell arborization (Figure 4A-B). With the use of this microfluidic device, we indeed make sure that we are analyzing axonal branches and not dendritic arborization. While axotomy/regeneration experiments were performed on cells plated at higher density in the device. We have better clarified this difference in the methods section, as follows (page: 23, lines 21-22):

"For axon branching analysis, the initial density was 5×10^4 cells per cm^2 . For axotomy experiments, the initial density was 10^5 cells per cm^2 ".

4. More images and quantification is needed in Figure 7 to sufficiently be convinced of the result. For 7a, please do biological replicates (three separate wells of each condition, for example) and perform the statistical analysis on this; technical replicates on a single sample of each conditions are not sufficient. For 7B, please show traces of representative axons in the experiment. For 7C, why do you not see the robust regeneration phenotypes shown in previous figures in the top condition? Images are not convincing in this case. I would suggest that authors quantify three biological replicates and show quantification as in prior figures.

Figure 7 has become figure 8 in the revised manuscript.

Fig.8A: As also requested by reviewer #1, minor point 3, this experiment has been now repeated to provide statistical significance (n=3 biological replicates).

Fig.8B: traces of representative axons have been included in the revised figure.

Fig.8D-E: a similar point has been raised by reviewer #2 (point 2). We have now provided quantification of the axon length in 17 independent transfections over 2 differentiation experiments. As the referee correctly noticed, axons are on average longer in regenerating untransfected FUS-P525L MNs (Figure 4C, average length 600um) than in non-targeting siRNA-transfected FUS-P525L MNs (Figure 8E, average length 500um). This difference could be due to stress induced by transfection. In any case, the comparison between non-targeting and NRN1 siRNAs shows that knockdown of NRN1 leads to highly significant impairment of regeneration (Figure 8E).

5. The model fits with the authors' data and I believe that the authors have provided good support for most of the model. The one piece where I'm a bit unsure is the part where FMRP and FUS compete for the same site on the 3' UTR. The luciferase assay provided by the authors is a solid start and, when combined with the FMRP RIP assay, suggests their model is likely. A more convincing demonstration of this part of the mechanism would help to strengthen the paper. The authors say that the HuD 3'UTR contains multiple possible regulatory sites. For example, could you delete the predicted binding site and eliminate both FMRP RIP and FUS binding? Alternatively, could you test whether FUS variants can be IP'ed for the RIP assay and evaluated for HuD binding in the presence or absence of FMRP?

We now provide this important evidence, as also requested by the other reviewers (Ref. 1 point 1 and Ref. 2 point 1). Specifically, we have performed an in vitro competition assay with in vitro transcribed RNA corresponding to 3 regions of the HuD 3'UTR. We found that the purified FUS-P525L protein competes with FMRP on one of these regions (comparing FUS-P525L with a mutant derivative in which RNA binding is impaired; see Daigle et al., 2013). These results have been included in the new Figure 2 (note that the old Figure 2 has been split in new Figures 2 and 3 to accommodate this new data).

6. I am left after reading the paper with a bit of disorientation about how this is all connected to ALS. It needs to be made more clear, with citations, why exuberant regeneration could contribute to ALS. It's also not entirely clear what is the evidence for extra axon branching in patients with this mutation in the first place? In doing some reading, the 525L mutation is associated with severe, juvenile ALS variants. The authors note that axon branching increases and outgrowth increases are common across many ALS causing mutations. Can the authors compare (literature is fine) the effect on primary outgrowth, or regenerative outgrowth, among different reports studying different ALS mutations? It would help to put these findings into an appropriate context.

The connection between aberrant axon branching and ALS has been recently reviewed (see ref. 24). In short, aberrant (meaning "increased") branching and growth has been observed in multiple models (zebrafish, mouse, human iPS) with a several individual mutations in ALS linked genes (SOD1, TDP-43, FUS, CCFN) or in the SMA gene SMN. We have referenced these works throughout the paper (refs. 25, 26, 27, 28, 29). We agree with the reviewer that comparison of these studies is of great interest, but we believe that it could be more suitable as the subject of a review article. The main message of our paper is that at the basis of these phenotypes, at least in the case of FUS, there is a dysregulated cross-regulative mechanism which involves 3 RBPs (FUS, FMRP and HuD), ultimately leading to increased HuD levels. Notably, while our paper was under revision, the Perrone-Bizzozero and Cereda groups reported increased HuD protein and RNA binding activity (and, interestingly, regulation of SOD1) in sporadic ALS patients (specifically in the motor cortex and not in the cerebellum) (ref. 72). We have mentioned this relevant finding in the revised paper as follows (page 14, lines 25-26; and page 15, line 1):

“Notably, recent evidence of HuD upregulation and increased binding activity in sporadic ALS patients’ motor cortex [72] suggests that the present findings might extend beyond FUS-ALS.”
Despite it is still unknown what causes HuD upregulation in sporadic ALS, this recent publication opens the possibility that our findings on HuD gain-of-function in FUS models extend beyond FUS as a more general feature in ALS.

Reviewers' comments:

Reviewer #1 (Remarks to the Author):

Although this manuscript has been extensively revised with providing additional data, the revised manuscript does not fully address the reviewer's major concerns. Particularly, a major experiment – the competition binding assay for FUS and FMRP – was not properly conducted, and some of the data lacks replicates with statistical significance, rendering the reviewer difficult to be convinced by the conclusions drawn by the authors.

1. Figure 2E shows the data for the FUS and FMRP competition binding assay on HuD 3' UTR. However, it is not clearly described how the FUS and FMRP competition binding assay on HuD 3' UTR was conducted both in the Methods section and the Figure Legend section. In the Methods section, it is indicated that the FUS P525L complex was immunoprecipitated from FUS-overexpressing cell line, in which other accessory proteins may be pulled down together with the mutant FUS (possibly including FMRP), and this is compared with RNA binding deficient mutant. What I can infer from this data is that the FUS complex binds to the Fragment 1 and 2 of HuD 3' UTR, but can not infer whether FMRP is binding and competing with the FUS complex for the same site.

Usually, protein-protein competition binding assay is conducted using purified recombinant proteins from bacteria. For example, in the presence of a fixed amount of HuD 3'UTR and FMRP, increasing amounts of FUS can be added to determine whether FMRP is competed by FUS. For example, please see this paper in Figure 5: rbFOX1/MBNL1 competition for CCUG RNA repeats binding contributes to myotonic dystrophy type 1/type 2 differences | Nature Communications

2. Most cell biology experiments ask for the average and SEM from at least three independent experiments. It is worrisome whether the experiments have been repeated properly. For example, Figure legends indicate that Figure 1C and Figure 1E have been conducted only twice. If these experiments were conducted three times, I would expect to see only three dots for each sample, each dot representing the average of many cells (>100 cells per sample would be ideal). I wonder what each data point represents in these figures.

3. The authors provide several bold statements without providing statistical significance in Figure 2B. For example, "HuD mRNA levels were reduced in FMRP immunoprecipitated samples from FUS P525L MNs (Page 6, Lines 14-15)". Please indicate whether the levels of HuD mRNAs or MAP1B mRNAs immunoprecipitated by FMRP are statistically different between FUS WT and P525L in Figure 2B.

In addition, Page 10, lines 11-12 is a bold statement, "Collectively these data point to increased levels of HuD targets in FUS mutant MNs as a consequence of the disruption of the FMRP-mediated negative regulation of HuD by FUS P525L". The authors would need to determine whether FMRP is less bound to HuD 3'UTR in FUS P525L compared to wild-type FUS. Figure 2A-B demonstrates this and would require statistical significance for Figure 2B.

4. GAP43 was introduced for the first time in Page 9, line 10. Is it known that GAP43 plays a role in the growth cones?

5. Quantification data is missing for Figure 7D.

6. Please revise Figure legends section.

- 1) In Figure 1C, what does third panel FUSWT + HuD represent? How this experiment was performed?
- 2) In Figure 1D, what is ANISO? Why was ANISO added to FUS P525L?
- 3) In Figure 2B, "MAP1" should be labeled as "MAP1B".
- 4) It is difficult to understand the data from the Figure Legends. Please revise Figure legends to clearly convey the information to the readers. For example, Figures 1C, D, E shows many data points, but are these data points from "2 independent differentiation experiments"? What does each data point indicate?

7. Extensive editing is required for the future audience to clearly understand the important findings from this paper.

- 1) In the abstract, each word for "Amyotrophic Lateral Sclerosis" or "Pluripotent Stem Cells" does not need to be capitalized.
- 2) Please revise the first paragraph of the Introduction (Page 3, lines 2-12) as this paper focuses on FUS gene. One option would be to focus on giving background on "FUS" gene and its function and remove all the contents related to TDP-43.
- 3) Please revise the third paragraph of the Introduction (Page 4, lines 2-10) because this paragraph makes the reader confused. The first line defines ALS feature as "axonal degeneration", but the rest of the sentences summarize the past literature in "axonal regeneration" or "increased axonal branching and growth". It would be helpful for the reader to understand how axonal "regeneration" at early disease stage contributes to "degeneration" at later disease stage and why early axonal regeneration would need to be studied in ALS. In addition, it would be great if the authors can clearly define the question that they are trying to ask and answer in this paper.

Reviewer #2 (Remarks to the Author):

The authors have substantially improved the manuscript. It is nice development for understanding the intricacies of RNA binding protein interactions and competitions. My concerns on the conclusions not being supported by the data are addressed by the new data included here. I have two minor issues that can be addressed by modest edits of the text.

1. In the introduction, the authors refer to HuD as a neural 'translation enhancer' – this could lead the reader to assume it only functions by enhancing translation, when it has other effects on RNAs. I think it is best referred to as a multi-functional RNA binding protein, or the authors should include its other functions with this descriptor.

2. The authors refer to NRN1 as a 'neurotrophic factor'. The protein certainly has growth promoting functions, for axons, dendrites, and synapses, but it does not fit with the classic concept of a neurotrophic factor like BDNF or GDNF. Despite a few publications referring to it as such, Nrn1 was initially identified as a plasticity gene (CPG15) whose expression is responsive to neurotrophic factors. I think it is best referred to as a growth promoting protein to avoid confusion with secreted neurotrophic factors.

Reviewer #3 (Remarks to the Author):

The additional data, pictures, and biological replicates provided by the authors were done appropriately. Figure 2 binding competition assay is very nice and strengthens the conclusions of the authors on this study. I am satisfied with the work of the authors and find that their studies adequately support their conclusions.

Minor comments:

- Please update abstract to fix grammar: “affect the activity of two other...”
- It is still confusing to readers when words like “changed” and “aberrant” and “alteration” are not qualified with a direction (up/down, promoted/inhibited, increased/decreased). I appreciate that the authors did modify their language a bit in the resubmission but more is needed for such a mechanistic study, so that readers can follow the logic more easily.
- Please clarify in methods regarding RIP: when the “artificial spike RNA” was added, and the operations used to adjust values based on its value. Also please note what this target was. Is this the same thing as mentioned as the “internal calibrator” in the methods section “real-time qRT-PCR”? If so, please unify your terminology throughout the methods.

Garone et al.
Point-by-point response to the referees' comments (Round 2)

Reviewers' comments:

Reviewer #1 (Remarks to the Author):

Although this manuscript has been extensively revised with providing additional data, the revised manuscript does not fully address the reviewer's major concerns. Particularly, a major experiment – the competition binding assay for FUS and FMRP – was not properly conducted, and some of the data lacks replicates with statistical significance, rendering the reviewer difficult to be convinced by the conclusions drawn by the authors.

In the revised version of the paper, we have included additional data and controls which, we hope, will fully address the concerns of the referee. In particular, we provide new evidence on the purification of the recombinant FUS proteins from cell extracts, used in the *in vitro* binding assay. Moreover, where required, we have strengthened the statistical analysis with additional replicates to support conclusions. In all other cases, we have better clarified that all experiments have been repeated at least 3 times. All other minor points and text edits requests raised by this referee have been also addressed, as detailed below.

1. Figure 2E shows the data for the FUS and FMRP competition binding assay on HuD 3' UTR. However, it is not clearly described how the FUS and FMRP competition binding assay on HuD 3' UTR was conducted both in the Methods section and the Figure Legend section. In the Methods section, it is indicated that the FUS P525L complex was immunoprecipitated from FUS-overexpressing cell line, in which other accessory proteins may be pulled down together with the mutant FUS (possibly including FMRP), and this is compared with RNA binding deficient mutant. What I can infer from this data is that the FUS complex binds to the Fragment 1 and 2 of HuD 3' UTR, but can not infer whether FMRP is binding and competing with the FUS complex for the same site. Usually, protein-protein competition binding assay is conducted using purified recombinant proteins from bacteria. For example, in the presence of a fixed amount of HuD 3'UTR and FMRP, increasing amounts of FUS can be added to determine whether FMRP is competed by FUS. For example, please see this paper in Figure 5: rbFOX1/MBNL1 competition for CCUG RNA repeats binding contributes to myotonic dystrophy type 1/type 2 differences | Nature Communications
We have more clearly explained this assay and provided additional experimental details in both the figure legend and in the methods section, as follows:

Figure legend: "(C) Schematic representation of the HuD transcript. The 3 regions of the 3'UTR (F1, F2, F3) used for *in vitro* binding assays are shown. (D) The *in vitro* binding assay was performed by incubating biotinylated transcripts corresponding to *HuD* 3'UTR regions F1, F2 or F3, or a portion of the Renilla luciferase coding sequence used as negative control (Neg. C), with HeLa cytoplasmic extract, followed by pull-down with streptavidin-conjugated beads. Western blot analysis was then performed with anti-FMRP antibody to detect FMRP binding. Anti-GAPDH was used as negative control. Input: 10% of the pull-down input sample. On the right, histogram showing quantification from 3 independent experiments. Values were calculated as fraction of Input (Student's t-test; paired; two tails; **p < 0.01; ns: p > 0.05). (E) The *in vitro* FMRP binding assay was repeated in presence of purified recombinant FUS proteins. F1 and F2 biotinylated transcripts were incubated with HeLa extract and purified RFP-flag-FUS^{P525L} (indicated as P525L) or an RNA-binding deficient mutant derived from RFP-flag-FUS^{P525L} (indicated as P525L 4F-L).

Western blot analysis was performed after pull-down with streptavidin-conjugated beads with anti-FMRP, anti-flag or anti-GAPDH antibody. Input: 10% of the pull-down input sample. Bottom: histograms showing quantification from 3 independent experiments. Values were calculated as fraction of Input and normalized to P525L (Student's t-test; paired; two tails; *p < 0.05; **p < 0.01)."

Methods: "In vitro binding assay was performed using *in vitro* transcribed biotinylated RNA (corresponding to 3 regions spanning the *HuD* 3'UTR), HeLa extract (containing FMRP protein), and purified recombinant FUS proteins from HeLa, as follows" [...] "For competition assay between FMRP and FUS, Streptavidin MagneSphere paramagnetic particles (Promega) were first washed in EMSA buffer 1X four times. Beads were then incubated in EMSA 1X with 150 µg of E. Coli tRNA at RT for 10 minutes and resuspended in EMSA 1X with 100 U/ml RNase inhibitor and 1X protease inhibitor cocktail (Roche). The assay was then performed in presence of 250 ng of biotinylated transcript, 75 µg of HeLa cytoplasmic lysate and 30 ng of purified FUS protein. RNA-protein complexes were then incubated with streptavidin beads in a final volume of 150 µl. Binding was carried out at 4°C with on a rotating wheel for 1 hour. Beads were then washed in 1X EMSA for three times. Each sample was resuspended in 20 µl PBS with 1X NuPage LDS (Invitrogen) and 2 mM DTT. Complexes were analyzed by western blotting."

For what concerns the purification of the recombinant FUS proteins, anti-FLAG M2 Magnetic Beads incubated with HeLa extract have been washed in stringent conditions with high salt buffer. Coomassie blue staining of the gels (Suppl Fig S5A) already suggested a good degree of purification from HeLa extracts. In order to exclude contamination by FMRP, as per referee's concern, we have now performed western blot analysis using the purified recombinant FUS samples used in the binding assay (and, as control, the HeLa extracts from which the recombinant flag-FUS was purified). This new result is shown in Suppl Fig S5B-D. Even after overexposure of the filter beyond saturation of the FMRP signal from HeLa extract, we could not observe any FMRP signal in the purified recombinant FUS samples, confirming high degree of purification in these conditions.

Finally, the statistically significant difference in the pull down of FMRP, when we added purified RNA-binding competent/incompetent FUS^{P525L} to the reaction, demonstrates reduced *in vitro* binding of FMRP in presence of a FUS protein that can compete for the same RNA substrate (Figure 2E).

2. Most cell biology experiments ask for the average and SEM from at least three independent experiments. It is worrisome whether the experiments have been repeated properly. For example, Figure legends indicate that Figure 1C and Figure 1E have been conducted only twice. If these experiments were conducted three times, I would expect to see only three dots for each sample, each dot representing the average of many cells (>100 cells per sample would be ideal). I wonder what each data point represents in these figures.

We have included a third replicate for Figure 1C. The graph in Figure 1E was already showing 3 replicates, mislabeled as 2 in the figure legend. We apologize for this mistake which has been fixed. In these graphs each dot represents a cell, and the histograms indicate the number of dots per cell. We preferred to present these data in this way to show the degree of cell-to-cell variability, which can occur in iPSC-based studies. For the experiments of Figure 6E and 8B-E we have clarified the number of replicates. For example, in Figure 8E we have used FUS^{P525L} MNs derived from 2 differentiation experiments, each plated in 6 or 8 wells and then each well was independently transfected with siRNAs (non-targeting or NRN1), resulting in "7 independent

transfections of non-targeting or NRN1 siRNAs from 2 differentiation experiments”.

3. The authors provide several bold statements without providing statistical significance in Figure 2B. For example, “HuD mRNA levels were reduced in FMRP immunoprecipitated samples from FUS P525L MNs (Page 6, Lines 14-15)”. Please indicate whether the levels of HuD mRNAs or MAP1B mRNAs immunoprecipitated by FMRP are statistically different between FUS WT and P525L in Figure 2B.

In addition, Page 10, lines 11-12 is a bold statement, “Collectively these data point to increased levels of HuD targets in FUS mutant MNs as a consequence of the disruption of the FMRP-mediated negative regulation of HuD by FUS P525L”. The authors would need to determine whether FMRP is less bound to HuD 3’UTR in FUS P525L compared to wild-type FUS. Figure 2A-B demonstrates this and would require statistical significance for Figure 2B.

We agree with the reviewer on the importance of the RIP assay of Figure 2A-B to support the conclusions of this part of the work. In order to increase confidence on this result, we have added two biological replicates and performed statistical analysis, which showed statistical significance for HuD and MAP1B mRNAs between FUS WT and P525L.

4. GAP43 was introduced for the first time in Page 9, line 10. Is it known that GAP43 plays a role in the growth cones?

GAP43 (growth-associated protein 43-kDa) is indeed localized at the growth cones and has been extensively studied in the context of axonal growth and plasticity. Notably, GAP43 localization at the growth cone depends on the HuD target sequence on its 3’UTR (an ARE sequence) (see ref. [44]). We have mentioned it as follows:

“Since HuD binding is known to localize GAP43 at growth cones [44], we focused on these structures and found a striking difference: while GAP43 protein was undetectable in FUS^{WT} MNs, a clear punctate signal was present in the FUS^{P525L} mutant (Figure 7D and Supplementary Figure S7A).”

5. Quantification data is missing for Figure 7D.

Quantification of GAP43 signal intensity is now shown. To accommodate this graph, we have moved individual panels of the IF in Supplementary fig. 7, keeping in the main figure the merged panels only.

6. Please revise Figure legends section.

1) In Figure 1C, what does third panel FUS^{WT} + HuD represent? How this experiment was performed?

In the previous version of the paper, “FUS^{WT} + HuD” represented FUS^{WT} hiPSCs overexpressing HuD under the Synapsin 1 promoter (SYN1::HuD). Since this cell line is introduced later in the text (see figure 6), we recognize that mentioning it at this point of the paper can generate some confusion. Since analysis of *HuD* (together with *NRN1* and *GAP43*) transcript levels was anyway shown in Figure 6A, we have decided to remove the third panel of Figure 1C for the sake of readability.

2) In Figure 1D, what is ANISO? Why was ANISO added to FUS P525L?

Anisomycin (ANISO) is a eukaryotic protein synthesis inhibitor, used here to provide a negative control for the PURO-PLA experiment. This has been explained in the figure legend as follows: “In panel (D), FUS^{P525L} hiPSC-derived spinal MNs treated with the eukaryotic protein synthesis inhibitor anisomycin (ANISO) were used as negative control of the PURO-PLA.”

3) In Figure 2B, “MAP1” should be labeled as “MAP1B”.

Correct. This has been fixed.

4) It is difficult to understand the data from the Figure Legends. Please revise Figure legends to clearly convey the information to the readers. For example, Figures 1C, D, E shows many data points, but are these data points from “2 independent differentiation experiments”? What does each data point indicate?

In these graphs each dot represents a cell, and the histograms indicate the number of dots per cell. See also point 2, above.

7. Extensive editing is required for the future audience to clearly understand the important findings from this paper.

1) In the abstract, each word for “Amyotrophic Lateral Sclerosis” or “Pluripotent Stem Cells” does not need to be capitalized.

Done, also in the main text.

2) Please revise the first paragraph of the Introduction (Page 3, lines 2-12) as this paper focuses on FUS gene. One option would be to focus on giving background on “FUS” gene and its function and remove all the contents related to TDP-43.

We have removed the contents related to TDP-43 as requested.

3) Please revise the third paragraph of the Introduction (Page 4, lines 2-10) because this paragraph makes the reader confused. The first line defines ALS feature as “axonal degeneration”, but the rest of the sentences summarize the past literature in “axonal regeneration” or “increased axonal branching and growth”. It would be helpful for the reader to understand how axonal “regeneration” at early disease stage contributes to “degeneration” at later disease stage and why early axonal regeneration would need to be studied in ALS. In addition, it would be great if the authors can clearly define the question that they are trying to ask and answer in this paper.

The possible contribution of early axonal alteration, including increased branching and axonal outgrowth, has been recently reviewed (ref. [24]). We have better clarified this point as follows: “Despite the underlying pathological mechanisms have not been yet fully elucidated, it has been proposed that axonal alteration, including aberrantly increased branching, can act as a trigger [24].”

Notably, increased axonal outgrowth and regeneration have been observed other ALS models, including a SOD1 mouse model (ref [29]).

Finally, we have better clarified the question of the paper in the last paragraph of the introduction as follows:

“In this work we aimed to gain insight into the molecular mechanisms leading to HuD upregulation in FUS mutant genetic background and into the functional consequences of HuD increase in MNs.”

Reviewer #2 (Remarks to the Author):

The authors have substantially improved the manuscript. It is nice development for understanding the intricacies of RNA binding protein interactions and competitions. My concerns on the

conclusions not being supported by the data are addressed by the new data included here. I have two minor issues that can be addressed by modest edits of the text.

We appreciate that the reviewer's concerns have been fully addressed by our new data. We have changed the text according to his/her minor points as specified below.

1. In the introduction, the authors refer to HuD as a neural 'translation enhancer' – this could lead the reader to assume it only functions by enhancing translation, when it has other effects on RNAs. I think it is best referred to as a multi-functional RNA binding protein, or the authors should include its other functions with this descriptor.

We have changed the text as follows:

"HuD is a neural multi-functional RBP and its overexpression induces increased neurite outgrowth in neuronal cell lines and primary neural progenitor cells [18-20]"

2. The authors refer to NRN1 as a 'neurotrophic factor'. The protein certainly has growth promoting functions, for axons, dendrites, and synapses, but it does not fit with the classic concept of a neurotrophic factor like BDNF or GDNF. Despite a few publications referring to it as such, Nrn1 was initially identified as a plasticity gene (CPG15) whose expression is responsive to neurotrophic factors. I think it is best referred to as a growth promoting protein to avoid confusion with secreted neurotrophic factors.

We now refer to NRN1 as a growth promoting protein as follows:

"HuD stabilizes the mRNA encoding the growth promoting protein NRN1 (Neuritin1) by binding its 3'UTR [41]"

"Increased levels of NRN1 in FUS mutant MNs prompted us to explore the possibility that the phenotypes described in Figures 4 and 5 are a direct consequence of aberrant activation of this growth promoting protein."

"The growth promoting protein NRN1 is one of the primary HuD targets in MNs. HuD stabilizes NRN1 mRNA via AU-rich element (ARE) binding on its 3'UTR [41,59]."

Reviewer #3 (Remarks to the Author):

The additional data, pictures, and biological replicates provided by the authors were done appropriately. Figure 2 binding competition assay is very nice and strengthens the conclusions of the authors on this study. I am satisfied with the work of the authors and find that their studies adequately support their conclusions.

We appreciate that the revision work has satisfied this reviewer. We have changed the text according to his/her minor comments as specified below.

Minor comments:

- Please update abstract to fix grammar: "affect the activity of two other..."

We have changed the text as follows:

"Using both human induced pluripotent stem cells and mouse models, we found that FUS-ALS causative mutations affect the activity of two relevant RBPs with important roles in neuronal RNA metabolism: HuD/ELAVL4 and FMRP."

- It is still confusing to readers when words like "changed" and "aberrant" and "alteration" are not

qualified with a direction (up/down, promoted/inhibited, increased/decreased). I appreciate that the authors did modify their language a bit in the resubmission but more is needed for such a mechanistic study, so that readers can follow the logic more easily.

We have better specified the direction of changes throughout the text, as follows:

Abstract: “Since similar phenotypes have been previously described in SOD1 and TDP-43 mutant models, increased axonal growth and branching might represent broad early events in the pathogenesis of ALS.”

Introduction: “Here we provide a mechanistic link between increased axon branching and growth upon axotomy and alteration of a cross-regulatory circuitry involving three RBPs: FUS, HuD and the fragile X mental retardation protein (FMRP).”

Introduction: “...but whether increased levels of HuD have functional consequences in *FUS* mutant MNs remains unexplored.”

Results: “Increased NRN1 and GAP43 levels in *FUS* mutant motoneurons. We next focused on downstream HuD targets, which could be altered as a consequence of increased levels of this RBP in *FUS* mutant MNs, and that might be involved in the observed axon phenotypes.”

Results: “Increased axon branching and growth upon axotomy in *FUS* mutant motoneurons are due to NRN1 upregulation”

Discussion: “Here we propose a regulatory mechanism for *HuD* translation in normal MNs and its increase in ALS.”

Discussion: “This phenotype is remarkably similar to increased neurite outgrowth and branching observed in motor neurons expressing the ALS SOD1-G93A mutant [29].”

- Please clarify in methods regarding RIP: when the “artificial spike RNA” was added, and the operations used to adjust values based on its value. Also please note what this target was. Is this the same thing as mentioned as the “internal calibrator” in the methods section “real-time qRT-PCR”? If so, please unify your terminology throughout the methods.

The spike RNA is not the “internal calibrator” (an endogenous transcript used for normalization of qRT-PCR) mentioned elsewhere, but an artificial RNA added to the RIP RNA sample for normalization. We have clarified its nature, when it was added, and the calculations to adjust values, as follows:

“An artificial spike RNA, i.e. an in vitro transcribed RNA fragment derived from the pcDNA3.1 plasmid, was added to the RNA fraction, which was then lysed with 250 μ l of TRIzol (Invitrogen) and extracted according to manufacturer’s instructions. RNA was analyzed by real-time qRT-PCR with iTaq Universal SYBR Green Supermix (Bio-Rad Laboratories). RIP data analysis was performed as follows. For each target, the mean Ct value from a technical duplicate was normalized to the input RNA fraction (at a 1/10 dilution) Ct value (Δ Ct) to account for RNA sample preparation differences, using the equation: Δ Ct = [Ct_{RIP or IgG} – Ct_{input} – Log₂10]. The percentage of input was calculated by linear conversion of the normalized Δ Ct as $2^{-\Delta$ Ct}. This value was then adjusted to take into account the difference in the amplification between the control IgG and the IP fractions using the artificial spike RNA, as follows. We first calculated the spike RNA Δ Ct and percentage of input as described above. Then, we normalized the percentage of input of each target using the percentage of input of the spike RNA.”

REVIEWERS' COMMENTS:

Reviewer #1 (Remarks to the Author):

All previously raised concerns by this reviewer have been addressed.